# Microbiology and Biochemistry of Pesticides Biodegradation

**DOI:** 10.3390/ijms242115969

**Published:** 2023-11-04

**Authors:** José Roberto Guerrero Ramírez, Lizbeth Alejandra Ibarra Muñoz, Nagamani Balagurusamy, José Ernesto Frías Ramírez, Leticia Alfaro Hernández, Javier Carrillo Campos

**Affiliations:** 1Instituto Tecnológico de Torreón, Tecnológico Nacional de México, Torreon 27170, Coahuila, Mexico; roberguermz95@gmail.com (J.R.G.R.); jfriasra@hotmail.com (J.E.F.R.); letyalf@yahoo.com.mx (L.A.H.); 2Laboratorio de Biorremediación, Facultad de Ciencias Biológicas, Universidad Autónoma de Coahuila, Torreon 27275, Coahuila, Mexico; ibarral@uadec.edu.mx (L.A.I.M.); bnagas@gmail.com (N.B.); 3Facultad de Zootecnia y Ecología, Universidad Autónoma de Chihuahua, Chihuahua 31453, Chihuahua, Mexico

**Keywords:** biodegradation, bioremediation, pesticides, microorganism, biodegradation pathways, degradation mechanism, degrading bacteria, biodegradation metabolism, microorganism degraders

## Abstract

Pesticides are chemicals used in agriculture, forestry, and, to some extent, public health. As effective as they can be, due to the limited biodegradability and toxicity of some of them, they can also have negative environmental and health impacts. Pesticide biodegradation is important because it can help mitigate the negative effects of pesticides. Many types of microorganisms, including bacteria, fungi, and algae, can degrade pesticides; microorganisms are able to bioremediate pesticides using diverse metabolic pathways where enzymatic degradation plays a crucial role in achieving chemical transformation of the pesticides. The growing concern about the environmental and health impacts of pesticides is pushing the industry of these products to develop more sustainable alternatives, such as high biodegradable chemicals. The degradative properties of microorganisms could be fully exploited using the advances in genetic engineering and biotechnology, paving the way for more effective bioremediation strategies, new technologies, and novel applications. The purpose of the current review is to discuss the microorganisms that have demonstrated their capacity to degrade pesticides and those categorized by the World Health Organization as important for the impact they may have on human health. A comprehensive list of microorganisms is presented, and some metabolic pathways and enzymes for pesticide degradation and the genetics behind this process are discussed. Due to the high number of microorganisms known to be capable of degrading pesticides and the low number of metabolic pathways that are fully described for this purpose, more research must be conducted in this field, and more enzymes and genes are yet to be discovered with the possibility of finding more efficient metabolic pathways for pesticide biodegradation.

## 1. Introduction

Currently, there are 5000 mega hectares (38% of the Earth’s surface) of land on Earth that have been recorded as being used for agricultural development. This surface can be divided into three parts, where one part represents crops, and the remaining two are prairies and pastures [1]. To keep crops suitable for human consumption, free of pests, and maximize their production, it is necessary to apply pesticides; according to Clark and Tilman [2], by 2030, the world population is expected to reach 8.5 billion people, which in turn will increase the use of pesticides.

A pesticide is any substance that can destroy, diminish, prevent, repel, control, attract, or even kill a pest or non-target organism; pesticides are classified by the World Health Organization (WHO) according to their degree of danger based on the lethal dose (LD_50_) in rats. Exposure to the substance, either by single or multiple exposures during a short period, causes an effect on the person who handles the product [3,4].

Due to the negative impact that pesticides have on the environment and non-target species, there is a constant search to find ways to reduce the effects caused by these compounds. Microorganisms have a considerable biochemical versatility, which they use to adapt and develop in different environments. This feature and others make microorganisms a suitable tool for the remediation of both soil and water. Additionally, their use is commonly less expensive compared to physical and chemical methods [5]. The great variety of microorganisms that have the capacity to degrade pesticides include fungi, bacteria, actinomycetes, and algae; these organisms can use xenobiotic compounds as a source of nutrients [6]. These microorganisms can use xenobiotic compounds as a source of nutrients through the use of enzymes [7].

An emphasis on bacteria has been present throughout previous decades; this is understandable since bacteria are more easily cultivated. Other microorganisms with a more complex and diverse metabolism, such as fungi, have also been studied but not as in-depth as bacteria. For instance, most of the metabolic pathways discussed in this review were studied from bacteria; this presents an opportunity for further research to focus on fungi or algae to explore if more efficient metabolic pathways for pesticide degradation are present in these microorganisms.

In this review, we will provide a comprehensive list of microorganisms that can degrade diverse pesticides, the metabolic pathways some of them use to achieve this process, and the genetics behind these metabolic pathways. Finally, a current panorama and perspectives on the application of microorganisms as a method for the bioremediation of xenobiotic compounds focused on agricultural pesticides will be presented.

## 2. Pesticides Used in Agriculture

According to the FAO (Food and Agriculture Organization) [8], a pesticide is any substance or mixture of substances intended for preventing, destroying, or controlling any pest, including vectors of human or animal disease, unwanted species of plants or animals causing harm during or otherwise interfering with the production, processing, storage, transport or marketing of food, agricultural commodities, wood, wood products, or animal feedstuffs, or substances which may be administered to animals for the control of insects, arachnids, or other pests in or on their bodies. The term includes substances intended for use as a plant growth regulator, defoliant, desiccant, or agent for thinning fruit or preventing the premature fall of fruit, and substances applied to crops either before or after harvest to protect the commodity from deterioration during storage and transport.

Pesticides have been used in agriculture for at least 80 years, and the first pesticide to be created was dichloro-diphenyl-trichloroethane (DDT). Twenty years after its discovery, DTT was banned for its use in agriculture [9]. Since the DDT discovery, several other types of pesticides have entered the market, most of them claiming safe use. Nonetheless, there is still public concern about the risks to health that the use of pesticides may present.

The use of pesticides has been increasing for the last few decades. From 2016 to 2021, the use of different pesticides experienced steady growth (Figure 1); when analyzed from 1990 to 2016, the trend of more tons per year is clearer [10].

### 2.1. Classification

There are different forms of classification of pesticides; they can be classified based on their chemical composition or by their target [10]. The WHO (World Health Organization) provides a classification based on the pesticide lethal dose 50 (LD_50_), which represents the dose required to kill half the tested population after a standardized test duration. WHO reports the LD_50_ for two types of exposition to the substances, dermal and oral [11]. This classification (Table 1) is very useful because it gives information about the health risks of any type of pesticide.

### 2.2. Extremely Hazardous Pesticides

According to WHO and the Globally Harmonized System of Classification and Labelling of Chemicals (GHS) of the UN (United Nations) [4], these types of pesticides may cause cancer and genetic defects and may have a negative impact on fertility or an unborn child. Table 2 shows pesticides that are used in agriculture that are extremely hazardous.

Aldicarb is a highly toxic carbamate insecticide that is soluble in water due to its polarity; in both target and non-target organisms, this pesticide acts as a cholinesterase inhibitor. Though the inhibition is reversible, the effect may be acute depending on the amount of pesticide exposure to the organism; in non-target organisms such as humans, it may have a considerable effect on the central nervous system [15,16]. This pesticide has been found in the drinking water of some U.S. cities, such as New York, where it exceeded the permissible limit [17].

Terbufos (TBF), an organophosphate (OP) pesticide, is used as an insecticide and nematicide; its use is prohibited in the European Union, and in the United States, the EPA lists it as a restricted use pesticide (RUP), meaning that it can be used for agricultural purposes, but it should not be used in homes [18,19].

Methyl parathion is used as an insecticide, nematicide, and acaricide/organophosphate acaricide to control agricultural crop pests; its application on crops consumed by children is prohibited [19,20].

### 2.3. Highly Hazardous Pesticides

According to the WHO classification, a pesticide that falls into class Ib is one whose median lethal dose is within the values of 5–50 mg kg^−1^ body weight orally and 50–200 mg kg^−1^ dermally. Table 3 shows the pesticides in use that fall within this classification [4].

Cyfluthrin is a type II pyrethroid, which is frequently used in veterinary medicine and agriculture against pests [23]; the residual pesticide may end up in food consumed by humans, affecting their health. If present in soils, it can have a detrimental effect on the soil microorganisms [24].

Tefluthrin is a type I pyrethroid, which contains a cyclopropane carboxylic acid moiety linked to an aromatic alcohol [25]. Its action on pests results in paralysis and death; it is applied to lepidopteran and coleopteran pests [26].

Carbofuran is an N-methylcarbamate pesticide used to control crop insect and nematode pests [27]. As a cholinesterase inhibitor, it can cause harmful effects on the health of non-target organisms such as mammals. Due to this and other health effects, the use of carbofuran has been banned in some countries, but it has not stopped being used in developing countries [28].

### 2.4. Moderately Hazardous Pesticides

According to the WHO classification, a pesticide that falls into class II is one whose median lethal dose is within the values of 50–2000 mg kg^−1^ body weight orally and 200–2000 mg kg^−1^ dermally. Table 4 shows the pesticides in use that fall within this classification [4].

DDT is a moderately toxic pesticide with a half-life of 4–30 years. In addition to being toxic, it is a recalcitrant chemical with a complex degradation process [37]. In the 1960s, it was observed that DDT was present in all life forms in addition to water, air, and soil [38].

Lambda-cyhalothrin is a pesticide belonging to the pyrethroid class, developed in 1984 [39,40]. It is an acaricide insecticide [41]. Concentrations of this pesticide have been detected in mixtures with other types of pesticides, and relatively high doses (ranging from 10 to 100 ng of active ingredient per liter of water) can commonly be detected [41].

Permethrin was the first photostable synthetic pyrethroid developed in 1972 by Michael Elliot [39]. This pesticide is one of the most common pesticides that can be found in the environment. Permethrin is commonly used in agricultural and domestic applications, and traces of this pesticide have been found and reported in water bodies [42]. It belongs to the group of organophosphates commonly used as insecticides. It is one of the insecticides used after the prohibition of the use of organochlorines. Studies have shown that complete biodegradation of permethrin can take more than 15 years [39,43]. It is one of the pesticides included in the European Water Framework Directive (Directive 2000/60/EC), monitoring its persistence in water [44].

Dimethoate is an insecticide commonly used to kill insects and mites. It targets the central nervous system of pests [45]. It is also used to control fungal diseases of fruits, vegetables, and field crops [46].

2,4-dichlorophenoxyacetic acid (2,4-D) is a phenoxy herbicide that was originally synthesized in 1941 and was commercialized for the first time in 1945, and since then, it has been one of the lowest-cost herbicides sold in the USA [38]. Even though it is considered an excellent herbicide, it is a disruptor of the endocrine system of mammals, highly toxic to the liver and kidneys, and has been cataloged as a carcinogenic and mutagenic pesticide [47].

Dicamba is a post-emergence pesticide that is used in a variety of crops where glyphosate-resistant weeds have emerged [48]. Considerable exposition to this pesticide can have a toxic effect on the liver and cellular functions. When comparing the toxicity of this pesticide with that reported in the pesticide data sheet, a great difference can be observed [48]. Toxic effects on mammals, from embryotoxicity and teratogenicity to neurotoxicity, have been demonstrated [49].

Cyanazine, a herbicide belonging to the triazine group, is one of the most widely applied pesticides in the US crop. This pesticide is commonly found in water samples [50]. According to some studies, the concentration of this pesticide has reached up to 1300 µg L^−1^ in surface water and 3500 µg L^−1^ in groundwater [51].

### 2.5. Slightly Hazardous Pesticides

A pesticide that falls into class III has a median lethal dose of >2000 mg kg^−1^ body weight orally and >2000 mg kg^−1^ dermally. Table 5 shows pesticides in use that fall within this classification [4].

Glyphosate is one of the most used herbicides in the world; it is used to control the growth of weeds in fields. After its application, the pesticide remains in the soil, reaching groundwater, and as a potential contaminant, it may affect wildlife and humans [55]. It is a non-selective herbicide [39]. This pesticide was originally patented as a chelator in 1950 [56].

Atrazine is a synthetic herbicide of the triazine group. This herbicide is responsible for modifying the development of plants by affecting the process of enzyme production and photosynthesis [57].

Metolachlor belongs to the chloroacetanilide herbicides, which are very common to find when analyzing water samples. This herbicide was banned in 2006 [54]; when introduced to the market, it was sold as a product containing the enantiomer pair (R and S). It was developed for the control of grasses and weeds and is recorded to have been used on at least 70 crops [58].

### 2.6. Unlikely to Present Acute Hazard Pesticides

The pesticides that are classified in this group are those that do not present alarming toxicity since, to cause lethal toxicity, very high quantities are needed, exceeding 5 g of pesticide [4]. Table 6 shows the pesticide in this classification that is used in agriculture.

Trifluralin is a pesticide capable of depolymerizing microtubules or altering the concentration of calcium ions in the cell, causing interference in the mitotic division of cells. It affects the meristems and the subterranean tissues of the plant [60]. This herbicide is used in pre-emergence stages for weed control, mainly in wheat crops [61].

## 3. Biodegradation of Pesticides

### 3.1. Microbial Diversity

Some pesticides are extremely resistant to degradation, making them persistent contaminants in the environment. Moreover, most of them may be considered a health concern. Due to the chemical stability of the chemicals, new strategies, such as biodegradation, are needed to achieve the removal of these molecules from the environment [62]. As Matsumura et al. [63] rightly argue, one environmentally friendly method is the use of microorganisms with the capacity to biodegrade these contaminating compounds that are present in soil and water.

Microorganisms have a strong capacity to adapt to the constant changes in the environment they inhabit, for example, mutation or induction. Microorganisms will use different types of metabolisms to metabolize these xenobiotic compounds, which they can later use as a source of carbon, nitrogen, phosphorus, energy, etc. Microbial metabolism of the pesticides ends in one of two scenarios: the complete biodegradation of the molecules or the mineralization of it; in this case, most of the by-products are suitable for their re-entry into the environment [63,64].

It should be emphasized that pesticide biodegradation is less expensive than traditional methods, which makes it financially viable for companies wishing to implement it, and that the by-products produced are less or almost not harmful to the environment [64,65]. Within the microbial kingdom, bacteria, fungi, and algae are currently being investigated for their ability to degrade pesticides [66].

#### 3.1.1. Bacteria

Multiple genera of bacteria (Table 7) have the metabolic tools to metabolize pesticides. In this process, pesticide molecules may be used as nutrients or as electron donors; the metabolism rate will depend on several biotic and abiotic factors, such as temperature, water availability, nutrient availability, other microorganisms present, and physical disruption of soil by agricultural practices. The overall result of the bacterial pesticide biodegradation process is the conversion of a highly toxic substance to a less or even non-toxic product [65,67].

#### 3.1.2. Fungi

Because fungi show mycelial growth, they are more frequently used for bioremediation in soil than in water, and they have the property of producing extracellular enzymes in sufficient quantities to produce the enzymes needed for bioremediation [215]. It must be emphasized that the right factors must be in place for the fungi to achieve proper degradation of the xenobiotic compounds in the soil, pH, minerals present, and moisture, to name a few factors [65].

Basidiomycetes is one of the main soil bioremediating agents against xenobiotic compounds. They are characterized by a high degradation capacity, and their bioremediation strategy relies heavily on the production of extracellular ligninolytic enzymes, converting toxic compounds into sources of energy and nutrients, going from complex to simple compounds [216]; a list of pesticide-degrading fungi is shown in Table 8.

**Table 8 ijms-24-15969-t008:** Pesticide degrading fungi.

WHO Pesticide Classification	Pesticide (% of Biodegradation Rate)	Fungi	Reference
Extremely Hazardous	Aldicarb (40–50%) Terbufos (50–100%) Methyl parathion (80–100%)	*Ascochyta* sp. CBS 237.37 *Trametes versicolor* *Coriolus versicolor* NBRC 9791 *Bjerkandera adusta* 8258 *Pleurotus ostreatus* 7989 *Phanerochaete chrysosporium* 3641 *Fusarium* sp. *Yarrowia lipolytica* *Aspergillus niger* AN400 *Penicillium citrinum* DL4M3 *Penicillium citrinum* DL9M3 *Fusarium proliferatum* DL11A *Aspergillus sydowii* *Penicillium decaturense* *Aspergillus niger* MRU01 *Aspergillus niger* NCIM 563	[217,218,219,220,221,222,223,224,225,226]
Highly Hazardous	Cyfluthrin (10–70%) Carbofuran (96%)	*Ascochyta* sp. CBS 237.37 *Trichoderma viride* 2211 *Aspergillus niger* ZD11 *Aspergillus nidulans* var. *dentatus* *Sepedonium maheswarium* *Trametes versicolor* *Mucor ramannianus* *Pichia anomala* *Trametes versicolor*	[217,227,228,229,230,231,232,233]
Moderately Hazardous	DDT (50–100%) Lambda- Cyhalothrin (50–100%) Permethrin (90%) Chlorpyrifos (40–100%) Dimethoate (60–97%) 2,4-D (2–30%)	*Ganoderma lingzhi* *Fomitopsis pinicola* *Gloeophyllum trabeum* *Cladosporium* sp. *Aspergillus sydowii* *Trichoderma* sp. *Cladosporium cladosporioides* Hu-01 *Rhodotorula glutinis* *Rhodotorula rubra* *Phanerochaete chrysosporium* *Trichoderma harzianum* *Trichoderma virens* *Byssochlamys spectabilis* *Aspergillus fumigates* *Aspergillus terreus TF1* *Verticillium* sp. *Aspergillus* sp. *Trichoderma viride* *Trichoderma harzianum* *Aspergillus niger* *Aspergillus oryzae* *Penicillium citrinum* *Aspergillus fumigates* *Trametes versicolor* *Penicillium chrysogenum* *Aspergillus niger* MRU01 *Eurotim* sp. F4 *Emericella* sp. F5 *Trichosporon* sp. *Penicillium implicatum* *Aspergillus viridinutans*	[96,120,137,216,225,234,235,236,237,238,239,240,241,242,243,244,245,246,247,248,249]
Slightly Hazardous	Glyphosate (60–80%) Atrazine (70–100%) Metolachlor (30%)	*Aspergillus flavus* *Aspergillus terricola* *Fusarium* sp. *Aspergillus niger* *Scopulariopsis* sp. *Trichoderma harzianum* *Fusarium oxysporum* *Penicillium notanum* *Aspergillus oryzae* A-F02 *Penicillium chrysogenum* *Fusarium dimerum* *Fusarium verticillioides* *Aspergillus fumigatus* *Penicillium citrinum* *Purpureocillium lilacinum* *Mucor* spp. *Sterilia* spp. *Trametes maxima* *Paecilomyces carneus* *Pleurotus ostreatus* INCQS 40310 *Trametes versicolor* *Bjerkandera adusta* *Pluteus cubensis* SXS320 *Gloelophyllum striatum* MCA7 *Agaricales* MCA17 *Polyporus* sp. MCA128 *Datronia stereoides* MCA167 *Datronia caperata* MCA5 *Metarhizium robertsii* *Trichoderma* sp. *Aspergillus* section *Flavi* *Pichia kudriavzevii* Atz-EN-01 *Penicillium* sp. yz11-22N2 *Saccharomyces cerevisiae* *Anthracophyllum discolor* *Glomus caledonium* *Aspergillus niger* *Candida xestobii* *Mortierella* *Kernia* *Chaetomium* *Trichosporon* *Candida tropicalis* *Penicillium oxalicum* MET-F-1	[165,167,250,251,252,253,254,255,256,257,258,259,260,261,262,263,264,265,266,267,268,269,270,271,272,273,274,275,276,277]
Unlikely Hazardous	Trifuralin (80%)	*Phanerochaete chrysosporium* *Trametes versicolor* *Penicillium simplicissimum* *Metacordyceps chlamydosporia* *Stachybotrys chartarum* *Alternia alternata*	[211,278]

#### 3.1.3. Algae

As mentioned by García-Galán [279], contrary to fungi and bacteria, algae can and should be cultured in aqueous media. A particularity of algae is that they can grow in low-quality water where other microorganisms would experience excessive stress, considerably hindering their ability to grow. Algae are good bioremediation agents of contaminated water, whether coming from the industrial, domestic, or agricultural sector; thanks to their metabolic versatility, algae can use effluents contaminated with heavy metals, pesticides (Table 9), or organic matter and transform them into a source of nitrogen (N), eliminating the excess of nitrogen in the environment. They can also sequester carbon and produce oxygen (O2) [280].

If metabolically impeded, algae can establish relationships with heterotrophic microorganisms to achieve the bioremediation of pesticides present in water [281,282].

**Table 9 ijms-24-15969-t009:** Some of the algae currently have a bioremediating function.

Algae	Degraded Pesticide	Reference
*Chlorococcum humicola* *Gracilaria verrucosa*	2,4-D ^1^	[283,284]
*Chlorella vulgaris* *Scenedesmus bijugatus*	Methyl Parathion ^1^	[285]
*Chlorococcum* sp.	DDT ^1^	[285]
*Selenastrum capricornutum* *Synechococcus elongatus* *Chlorella vulgaris* *Chlorella* sp.	Atrazine (60–80%)	[286,287,288]
*Oscillatoria limnetica* *Skeletonema costatum* *Emiliania huxleyi* *Isochrysis galbana*	Glyphosate ^1^	[289,290]
*Chlorella* sp. *Scenedesmus* sp.	Chlorpyrifos ^1^	[291]
*Chlorella vulgaris*	Carbofuran (100%)	[292]
*Chlorella vulgaris*	Dimethoate (100%)	[292]
*Chlorella vulgaris*	Metolachlor (100%)	[292]

^1^ Biodegradation rates for these pesticides are not reported.

#### 3.1.4. Actinomycetes

Actinomycetes have an established ability to metabolize xenobiotic chemicals from soil and water. They are capable of adapting to different environmental setups; for example, they can grow well in acidic and alkaline conditions; this is important because the availability of different toxic compounds could be determined by this chemical factor. The genus Streptomyces is one of the most researched members of the actinomycetes; they are saprophytic bacteria that can be found both in soil and water [7,293,294].

Actinobacteria, such as *Streptomyces*, can use pesticides as a carbon source, degrading inorganic compounds completely and rendering them non-toxic to the environment [293]. *Streptomycetes* can utilize several metabolic tools to achieve bioremediation processes, one of which is the production of enzymes such as hydrolases, glucosyltransferases, xylanases, laccases, and proteinases [7].

Using a consortium of different genera and species of bacteria makes bioremediation more feasible. *Streptomycetes* can degrade different families of pesticides, such as organochlorines, organophosphates, pyrethroids, and urea [7]. According to Alvarez [295,296], the most outstanding genera belonging to the actinobacteria are *Frankia*, *Janibacter*, *Kokuria*, *Mycobacterium*, *Nocardia*, *Rhodococcus*, *Arthrobacter*, *Pseudonocardia*, and *Streptomyces*.

### 3.2. Metabolic Pathways

Metabolic pathways play a crucial role in the biodegradation of pesticides. Microorganisms utilize various metabolic pathways to break down pesticides into less harmful compounds. These pathways may include mitochondrial energy metabolism, fatty acid and lipid metabolism, amino acid metabolism, oxidative and hydrolytic pathways, and methylation.

Understanding these pathways is essential for two reasons. First, for developing safe and efficient pesticide use and bioremediation strategies for contaminated soil and water. Second, a comprehensive understanding of the enzymes involved in the metabolic pathways could allow the use of metabolic engineering or DNA recombinant techniques to use these biomolecules and, to a certain degree, not depend on the microorganisms.

The present section discusses metabolic pathways for the degradation of different pesticides. Almost all the information comes from bacteria; this observation is important because it exposes the need to study the metabolic pathways of pesticides from different microorganisms, such as fungi and algae.

Fungi exhibit a greater metabolic diversity compared to bacteria; this is evident in various aspects of their metabolic capabilities and interactions within ecosystems.

One key aspect is the diversity of fungal cytochrome P450 enzymes. Fungi possess a wider range of cytochrome P450 families than plants, animals, and bacteria [297]. These enzymes play a crucial role in the metabolism of various compounds, including xenobiotics and natural products [298]; the tremendous variation in fungal cytochrome P450s suggests that fungi have evolved diverse metabolic functions to meet novel metabolic needs [297].

Furthermore, studies have shown that fungi have a greater metabolic activity and diversity in certain environments. For example, in forest soils, fungi are more important and active at low temperatures than bacteria [299]. In suboptimal combinations of temperature and moisture, the cultivable bacteria in planted soil exhibit higher activity and metabolic diversity compared to unplanted soil, while the cultivable fungi in planted soil exhibit higher metabolic diversity than those in unplanted soil [300]. These findings highlight the metabolic versatility of fungi in different environmental conditions.

The metabolic diversity of fungi also has implications for carbon turnover and nutrient cycling in ecosystems. Fungal decomposers have wider enzymatic capabilities than bacteria, allowing them to mineralize low-quality substrates like particulate leaf litter [301]. Fungi dominate over bacteria in terms of biomass, production, and enzymatic substrate degradation in freshwater ecosystems [302]. These findings highlight the importance of fungal metabolic diversity and how it can be exploited for the biodegradation of pesticides.

#### 3.2.1. Extremely Hazardous Pesticides

##### Aldicarb

The metabolic degradation of aldicarb from microorganisms, such as bacteria, proceeds by oxidative and hydrolytic pathways in which they use this carbamate as the only source of carbon and nitrogen; therefore, their growth depends on the use of aldicarb (Figure 2) [67,303].

The first phase in metabolic degradation is sulfur oxidation, where aldicarb is first oxidized to aldicarb sulfoxide (2-methyl-2-(methylsulfinyl) propionaldehyde O-(methylcarbamoyl) oxime) and subsequently to aldicarb sulfone (2-methyl-2-(methylsulfonyl) propionaldehyde O-(methylcarbamoyl) oxime) [303,304]; these intermediates may enhance crop production but will persist in soil with a toxicity similar to that of the original aldicarb [303,305].

In the second stage, hydrolysis of sulfone and sulfoxides occurs via the enzymatic action of aldicarb hydrolase. This enzyme was detected in a cell-free extract of *Enterobacter cloacae* strain TA7 in the biodegradation of carbamates [296,303]. The products of the enzyme reaction are carbamic acid, which breaks down into carbon dioxide (CO_2_) and a corresponding amine [15], as well as N-methyl-carbamic acid and oximes, which undergo dehydration to form nitriles [303].

The products derived from the hydrolytic route are less toxic than aldicarb. It has been reported that bacteria belonging to the genera *Arthrobacter*, *Acinetobacter*, *Enterobacter*, *Bacillus*, *Pseudomonas*, *Methylobacterium*, and *Kocuria* can use aldicarb and its degradation products in the form of nitrogen [304].

##### Terbufos

Terbufos (TBF) [S-t-butylthiomethyl-O,O-diethyl phosphorodithioate] is an organophosphate (OP) pesticide used as an insecticide and nematicide [18]. Terbufos is vulnerable to the enzymatic activity of organophosphorus hydrolase (OPH), an enzyme known to be very efficient in degrading organophosphorus compounds [306]. It is also subject to other microbial enzymatic reactions, such as hydrolysis, oxidation, alkylation, and dealkylation [307,308].

Biodegradation of terbufos requires several steps (Figure 3). The first step is the generation of the intermediate terbufos-oxon, which is formed by oxidative desulfurization by an -OH radical on the P-S bond. Then, the interfacial transfer of a single electron from the sulfur atom near the phosphorus atom leads to the formation of the cation radical terbufos and the cleavage of the C-S bond in this radical, causing the formation of (C2H5O)2P(S)S- radicals which are the precursor of the O,O-diethyl phosphorodithioic ester [309,310].

The intermediate tert-butyl-hydroxymethyl sulfide is hydrolytically formed by the C-S bond of terbufos and leads to the formation of -SC(CH3)3 radicals, which are precursors of tert-butanethiol; recombination of these radicals can form dimers such as di-tert-butyl disulfide [310].

Finally, cleavage of the C-S bond leads to the formation of the tert-butyl carbonium ion, and hydrolysis of this molecule produces tert-butanol, dehydrogenation leads to the formation of isobutene which is oxidized to form acetone [309].

##### Methyl Parathion

Methyl parathion is produced by the reaction of O,O-dimethyl phosphorochloridothionate and 4-nitrophenol sodium salt in an acetone solvent [311]. Organic phosphorus hydrolase (OPH) hydrolyzes methyl parathion into 4-nitrophenol (PNP) and dimethylphosphate (DMP); this hydrolytic reaction is the first step in the degradation of methyl parathion by soil microorganisms [311,312] (Figure 4). In addition, OPHs are important bacterial enzymes as they participate in the hydrolyzation of PO and P=S bonds [313].

PNP is hydrolyzed to benzoquinone, which is subsequently transformed into hydroquinone, γ-hydroxymuconic semialdehyde, and maleylacetate. These reactions are important in the degradation process of methyl parathion to finally form β-ketoadipate [314]. The enzyme monooxygenase is involved in catalyzing the reaction of PNP to benzoquinone in the presence of FAD and NADH [315].

#### 3.2.2. Highly Hazardous Pesticides

##### Cyfluthrin

Degradation of this pyrethroid begins with the transformation of the carboxylester bond by cleavage to yield 2,2,3,3-tetramethyl-cyclopropanomethanol and 4-fluoro-3-phenexy-benzoic acid [316] (Figure 5). This reaction is the primary step of biodegradation of Cyfluthrin [88]. Then, 4-fluoro-3-phenexy-benzoic acid undergoes diaryl cleavage, resulting in a molecule of 3,5-dimethoxy-phenol and a molecule of phenol; both are further metabolized [317].

Carboxylesterases are essential in the metabolism of various living organisms and are produced as a defense to metabolize pesticides and insecticides [318]. In addition, they efficiently hydrolyze cyfluthrin to its corresponding acids and alcohols to reduce toxicity [319].

##### Tefluthrin

The degradation process of tefluthrin, a chemical lacking the α-cyano group in the phenoxybenzyl moiety [320], includes hydrolysis of the central ester bond and oxidation at several points (Figure 6) [321].

After the first step of hydrolysis, two intermediates are formed, cyclopropane carboxylic acid and PBAlc; the latter is oxidized to PBAld, and finally, PBA1d is transformed to either 1,2-benzenedicarboxylic acid or 1,2-benzenedicarboxylic butyl dacyl ester [322].

Different enzymes are involved in the biodegradation process of tefluthrin, such as carboxylesterase, which is the most purified enzyme of pyrethroid-degrading microorganisms; the enzymes monooxygenase and aminopeptidase are attributed to the hydrolysis of the ester bond during microbial degradation [25].

Transformation of tefluthrin occurs through the involvement of carboxylesterases at the central ester bond or monooxygenases at one or more of the acid or alcohol binding sites [323]. Monooxygenases are mediated by cytochrome P450, a metabolic system involved in the metabolism of xenobiotics, such as insecticides, in all living organisms, including microorganisms and plants [324]. They also catalyze the degradation of aromatic compounds by introducing an oxygen molecule, which increases their reactivity and solubility [324,325].

##### Carbofuran

Biodegradation of carbofuran proceeds in three main steps: hydrolysis of the carbamate bond, processing the aromatic fraction, and subsequent degradation of the aromatic ring [85].

Hydrolysis of carbofuran involves the participation of a hydrolase to separate the ester bond of the carbonyl group of N-methylcarbamic acid attached to phenol and the amide bond of methylcarbamic acid to produce carbofuran-7-phenol (7-phenol (2,3-dihydro-2,2-dimethyl-7-benzofuranol)), CO_2_, and methylamine (Figure 7) [325,326].

Carbofuran-7-phenol is then converted to 3-(2-hydroxy-2-methylpropyl) benzene-1,2-diol by carbofuran-7-phenol hydrolase [327]. The enzyme carbofuran hydrolase is encoded by the *mcd* gene, which was cloned from the plasmid DNA of *Achromobacter* sp. WM111, and it was observed that some carbofuran-degrading bacteria have sequence homology with this gene [328]. Carbofuran-7-phenol is the main metabolite produced in this process, which is considered less toxic than the parent compound, while methylamine is used as a carbon source by carbofuran-degrading microorganisms [326].

It has been reported that some bacteria that degrade carbofuran into carbofuran phenol belong to the genera *Pseudomonas*, *Flavobacterium*, *Achromobacter*, *Sphingomonas*, *Novosphingobium*, and *Paracoccus* [27]. By meta-scission of the aromatic ring, 3-(2-hydroxy-2-methylpropyl) benzene-1,2-diol is formed. Oxidation of this intermediate produces 2,8-dihydroxy-8-methyl-6-oxonone-2,4-dienoic acid; hydroxylation of this product leads to the production of 3-hydroxy-3-methyl butanoic acid and 2-oxopent-4-enoate, which is then converted to acetyl-CoA and pyruvate [85,327].

#### 3.2.3. Moderately Hazardous Pesticides

##### DDT

DDT DDT (1,1,1-trichloro-2,2-bis(p-chlorophenyl) ethane) is first dechlorinated and transformed to either 1,1-dichloro-2,2-bis(p-chlorophenyl) ethane (DDD) or 2,2-bis(p-chlorophenyl)-1,1-dichloroethylene (DDE) [329]. Both intermediates are more toxic than the original molecule; DDD can also be transformed into DDE [330] (Figure 8).

Several chemical reactions occur in the degradation of DDT, but mainly reductive dechlorination, which occurs through the dechlorination of the aliphatic chloroethyl group of the molecule [331]. Other chemical reactions that may be involved during the degradation process are dehydrohalogenation, dioxygenation, hydroxylation, hydrogenation, and meta-ring cleavage [307]. In addition, there are enzymes involved in the process, such as dehydrochlorinase, dioxygenase, reductase, decarboxylase, and hydrolase [331].

As mentioned before, the first primary intermediate in the DDT metabolic biodegradation route is DDD or DDE. Each of them is produced under different growth conditions. DDD is more common in anaerobic conditions, while DDE is associated with aerobic conditions [332]. Under aerobic conditions, DDT is transformed into DDE by dehydrochlorination carried out by the dehydrochlorinase enzyme [333]. Only a few microorganisms can completely degrade DDE to CO_2_ using co-metabolism of biphenyl to obtain biphenyl dioxygenase, an enzyme required to degrade DDE [334].

DDE and DDD are transformed to DDMU (1-chloro-2,2-bis(4′-chlorophenyl) ethylene), which is transformed into DDOH (2,2-bis(p-chlorophenyl) ethanol) and DDA (bis(4′-chlorophenyl) acetate) by hydroxylation and carboxylation, respectively, and finally mineralized to carbon dioxide [37]. DDMU is hydroxylated to DDOH by a hydroxylase, which has been detected in some bacteria, such as *P. aeruginosa* [97]. DDA is obtained by the carboxylation of DDOH [98].

##### Lambda-Cyhalothrin

Pyrethroids are insecticides containing an ester bond formed by alcohol and an acid. Lambda-cyhalothrin belongs to the type II pyrethroids, in which an alpha-cyano group is present in the phenylbenzyl alcohol position [323,335]. The main mechanism of biodegradation of type II pyrethroids is the hydrolysis of their carboxyl ester bonds, in which metabolites, such as PBA (3-phenoxybenzoic acid), PBAlc (3-phenoxybenzyl alcohol) and PBAld (3-phenoxybenzaldehyde), are formed [336].

Hydrolysis of ester bonds is performed by a carboxylesterase. This type of enzyme plays a fundamental role in the detoxification of pyrethroids, and some genes of carboxylesterase enzymes involved in the degradation of pyrethroids have been identified [323].

After hydrolysis of the carboxyl ester bond, 2-hydroxy-2-(3-phenoxyphenyl) acetonitrile is formed, which is converted to PBAld (3-phenoxybenzamide), and both compounds can be transformed into PBA (Figure 9) [25]. In addition, PBAld can become PBAlc, while PBAlc becomes PBA or PBAld [337]. The catalytic conversion of PBAld to PBA involves aldehyde oxidizing enzymes, such as aldehyde dehydrogenase. PBAlc is oxidized to PBAld by an alcohol dehydrogenase [338]. The *aldh* gene encoding aldehyde dehydrogenase has been found to be activated by pesticide presence in *Bacillus* spp. [339].

##### Permethrin

Permethrin (Per) is a type I pyrethroid with no cyanide in its chemical composition, and it is present in two forms of diastereomers, cis-Per and trans-Per [340]. An important step in the degradation of permethrin is ester cleavage, which allows this process to produce its metabolites [317,341] (Figure 10). During biodegradation, through the action of carboxylesterase, the metabolites 3-phenoxybenzyl alcohol (PBAlc) and 3-phenoxybenzaldehyde (PBAld) [342] are obtained. In addition, cyclopropanic acid (Cl2CA) is produced, and the PBAlc fragment is often intermediate in the photocatabolism of permethrin that can be oxidized to 3-phenoxybenzoic acid (PBAcid) [343]. The decarboxylation of cyclopropanic acid and phenoxybenzoic acid allows the production of CO_2_ [342].

##### Chlorpyrifos

Chlorpyrifos (CP) is the common name for the insecticide 0,0-diethyl 0-(3,5,6-trichloro-2-pyridinyl)-phosphorothioate, which is commonly used in the treatment of crops, turf, and ornamentals [344]. The degradation pathway of this insecticide comprises different metabolic steps (Figure 11). In the first step, chlorpyrifos is converted to chlorpyrifos-oxon (CPO) by oxidative desulfurization performed by an oxidase enzyme [345].

Chlorpyrifos reacts with hydroxyl radicals produced photochemically in the atmosphere to enable the formation of CPOs. Chlorpyrifos-oxon (CPO) is an unstable intermediate formed from chlorpyrifos by oxidative desulfurization or acylation. CPO hydrolyzes rapidly to TCP and diethylphosphate (DTP) in alkaline soils [346]. After this, two other metabolites are produced: 3,5,6-trichloro-2-pyridinol (TCP) and diethyl thiophosphate (DETP) [347].

The hydrolysis of chlorpyrifos is important for degradation, producing 3,5,6-trichloro-2-methoxy pyridine (TMP) and deactivating CPO to TCP [345]. In addition, reductive dechlorination produces 2,3-dihydroxypyridine, which hydrolyzes to 2,5,6-trihydroxypyridine, and metabolites are subsequently oxidized to aliphatic amines, inorganic phosphate, carbon fragments, etc. 2,3-dihydroxypyridine can also be broken down to produce maleamic acid, which in turn is oxidized to pyruvic acid, finally entering the Krebs cycle [345]. Chlorpyrifos is ultimately converted to CO_2_, or its metabolites are integrated into organic soil matter.

DETP is hydrolyzed to phosphorothioic acid and ethanol, where it is subsequently used by CP-degrading microorganisms as a source of sulfur, phosphorus, and carbon [347]. Enzymes, such as hydrolase, phosphotriesterase, phosphatase, catalase, and oxidase, hydrolyze chlorpyrifos by cleavage of the P-O, P-F, and P-S bonds [348].

##### Dimethoate

Biodegradation of dimethoate is achieved mainly by bacteria; two main pathways of biodegradation have been documented, and their intermediate metabolites have been detected and confirmed. In the first pathway (Figure 12A), dimethoate is first oxidized to omethoate, and the result of the metabolic route is two molecules, Aspartylglycine ethyl ester and O, O, O-Trimethyl thiophosphate. Both are mineralized and assimilated by the cell. In this pathway, two types of enzymes are proposed to participate: phosphatase and amidase [93].

In the second pathway (Figure 12B), dimethoate is first oxidized to form dimethoate carboxylic acid by the release of a molecule of methylamine. Dimethoate carboxylic acid may be decarboxylated or oxidized, and there are two possible products of the metabolic route: O,O,O-trimethyl phosphoric ester and phosphorothioic O,O,S-acid [136]. Enzymes involved in this second pathway remain to be discovered.

##### 2,4-Dichlorophenoxyacetic Acid

2,4-D is usually referred to as the oldest organic herbicide; it is used against wide-leaf weeds in different crops, including rice, wheat, sorghum, sugar cane, and corn [349]. It is a molecule that mimics the action of auxins, promoting the synthesis of metabolites such as ethylene and ABA, triggering cell death. It has been used for more than 80 years [350], and due to its high environmental persistence, it can accumulate in soil and eventually contaminate underground water [349].

Biological degradation of 2,4-D has been documented in fungi and bacteria. In bacteria, two pathways have been characterized [351], and the enzymes that participate in both are known and well-studied (Figure 13). In both metabolic pathways, the enzymes involved are oxidoreductases, except for one dehalogenase that participates in the second pathway (Figure 13B).

Recently, an engineered strain of *E. coli* succeeded at degrading 2,4-D [352]. In both metabolic pathways, the result is a molecule that enters the Krebs cycle and can be used for the central metabolism of the cell.

In fungi, the full biodegradation pathway has not been elucidated, but there is evidence that the cytochrome P450 enzymes may be involved to some extent in the fungal metabolism of this herbicide [353].

##### Dicamba

Dicamba (2-methoxy-3,6-dichlorobenzoic acid) is an auxin mimic herbicide used to control wide-leaf weeds that are used in a variety of crops [354]. It is one of the most commonly used herbicides [355], and due to its chemical properties, the off-target movement of this herbicide considerably poses a risk for contamination of soil, ground, and surface water [356].

Biodegradation of Dicamba has been documented mainly in bacteria; the experimentally confirmed metabolic pathway involves the dechlorination and demethylation of the molecule to end up with a molecule of 2-chloromaleylpyruvate (Figure 14) [357]; subsequent enzymatic transformation of this molecule has been inferred from the homology analysis of the genes in the operon, where two or three more enzymes participate with a proposed final product of pyruvate and fumarate/maleate. Both of them can be incorporated into the central metabolism of the cell [357].

#### 3.2.4. Slightly Hazardous Pesticides

##### Glyphosate

Glyphosate is one of the most commonly used wide-spectrum herbicides. For more than 40 years, it has been used in a variety of crops; it is considered to be the number one herbicide used worldwide. The existence of genetically modified crops that are resistant to glyphosate has caused a surge in its use during the last 20 years.

Glyphosate acts as an inhibitor of the enzyme EnolPyruvylShikimate-3-Phosphate Synthase (EPSPS), causing the plant to not be able to synthesize aromatic amino acids and eventually causing cell death.

Its bioremediation is performed by bacteria and fungi [358]. In bacteria, two different metabolic pathways have been elucidated; the first one involves the dephosphorylation of the molecule, and oxidation of the metabolic intermediate results in a molecule of glycine and formaldehyde, both of which are used as carbon sources. In the second metabolic pathway, the molecule is first oxidized, resulting in a metabolic intermediate and a molecule of glyoxylate, which the bacteria can use in the glyoxylate cycle. The intermediate is also oxidized, and a molecule of formaldehyde is the resulting product (Figure 15) [359].

##### Atrazine

Atrazine is an s-triazine-derived herbicide that has different uses, such as agricultural applications in corn, sorghum, and sugarcane and non-agricultural applications in forestry and conifers [360]. Atrazine inhibits one subunit of the photosystem II in plants, halting this process and ultimately causing plant death.

Due to its chemical structure, aerobic degradation of the molecule is difficult, and for this reason, microbial degradation is usually performed by a consortium of microorganisms rather than just one microorganism doing the job [361]. Three pathways have been elucidated with a common product, cyanuric acid. The first pathway (Figure 16A) is the most common one, mainly found in bacteria, the second (Figure 16B) and third (Figure 16C) are usually associated with microorganisms consortia [360]. 

In the three pathways, the dealkylation and dechlorination of the molecule are performed by oxygenases and hydrolases. Cyanuric acid is degraded to ammonia and carbon dioxide and used as a source of nitrogen and carbon, respectively (Figure 16).

##### Metolachlor

Metolachlor is a pre-emergent herbicide that is used to control grasses and some weeds in corn, sorghum, soybean, and cotton [362]. This type of herbicide acts as an alkylating agent that can bind to different types of proteins within the plant, but the principal mechanism of action is the inhibition of lipid biosynthesis [362].

Metolachlor has a considerable half-life, and its persistence in the environment can cause its mobilization to water bodies. The biodegradation process of this molecule has been studied since 1990 [363]. Several metabolites of the metolachlor biodegradation pathway have been identified so far, but a complete metabolic pathway remains to be fully discovered; the general pathway that has been described involves the dichlorination (Figure 17B) or hydroxylation (Figure 17A) of the molecule. In both steps, intermediates are formed, which are then further metabolized to carbon dioxide [275] (Figure 17).

### 3.3. Genetics of Pesticide Biodegradation

Analysis of the genes involved in the biodegradation of pesticides is key to fully comprehending this biological process. Understanding the genetics could help to trace the evolutionary history of pesticide biodegradation; it could also support the engineering of microorganisms that can degrade pesticides more efficiently and the development of new bioremediation techniques that can be used to remove pesticides from contaminated soils, sediments, and water.

Genes related to the degradation of carbamate pesticides, such as carbofuran, have been identified in more than 50 carbofuran-degrading bacteria [364]; the *mcd* gene encodes a carbofuran hydrolase and was first identified in the pPDL11 plasmid. The gene for the degradation of carbaryl *cehA*, another carbamate pesticide, also encodes a hydrolase. Homology between these two genes is very low, and the carbaryl hydrolase has no activity in the presence of carbofuran [365]. A homolog of the *cehA* gene has been found in the bacteria *Novosphingoium* sp. KN65.2. The gene *cdfJ* encodes a hydrolase that shows enzymatic activity both with carbaryl and carbofuran [85]. Other genes for carbamate pesticide degradation have been found in different bacteria, and most of them share a high homology with the *cehA* gene [304]. All the genes are present in plasmids and are thought to be mobile elements that can be shared between bacteria.

Genes involved in the degradation of organophosphates pesticides, such as Terbufos, Methyl Parathion, Chlorpyrifos, Dimethoate, and Glyphosate were discovered in the late 80’s [366]. The *opd* gene encodes an organophosphorus hydrolase and was found in a plasmid. A chromosomal homolog of the *opd* gene, the *opdA* gene, was found in *Agrobacterium radiobacter* P230 [367]. Whole genome sequencing has allowed the discovery of several *opd*-like genes in different microorganisms, all of which are phylogenetically closely related, indicating horizontal mobility [315].

The *mpd* gene is also involved in the degradation of organophosphate pesticides, particularly parathion and methyl parathion; the gene encodes a hydrolase that has been well characterized. The hydrolase has conserved β-lactamase domains [368], and several homolog genes have been found in different microorganisms, although experimental evidence of their activity as parathion hydrolases is yet to be proved [368].

The *phnJ* gene encodes a C-P lyase that participates in the biodegradation process of glyphosate; the gene is part of the *phn* operon, and the C-P lyase has an important physiological function, possibly explaining that the gene is highly conserved within bacteria [369]. Glyphosate oxidation is performed by the enzyme glyphosate oxidase, which is the product of the gene *gox* [370]; the gene has not been fully studied, and whether it is present or not in other microbial genomes is unknown.

Pyrethroids such as Cyfluthrin, Tefluthrin, Lambda-cyhalothrin, and Permethrin are biodegraded by microorganisms by the action of carboxylesterases, also known as pyrethroid hydrolases. These enzymes and their genes have been identified in mammals, insects, and microorganisms [317]. Several bacterial pyrethroid degrading genes like *pytY*, *pytZ*, *estP*, *pytH*, and *pye* have been identified [37].

The genes involved in the degradation of organochloride pesticides like DDT have been described by genome annotation of *Stenotrophomonas* sp., *dhc,* and *rdh* genes are involved in the transformation of DDT to DDMU; *sds, dhg,* and *hdt* genes are involved in the transformation of DDMU to DDHO. *dlc* and *hdl* genes are involved in the last step of DDT biodegradation [371].

Genes involved in the biodegradation process of the chlorophenoxy herbicide Dicamba are situated in two different operons. In the first one, *dmt* genes encoding a demethylase are responsible for the first step in the biodegradation route of dicamba [372]; in the second operon, we found the genes *dsmABC*, *dtdA*, *dsmD*, *dsmG*, and *dsmE* that are responsible for the steps of reduction, oxidation, and dichlorination of the demethylated metabolite of dicamba. 2,4-D is another chlorophenoxy herbicide. The genes involved in both metabolic pathways have been identified; the *tfd* operon [373] comprising the genes *tfdA* and *tfdBCDEF_(II)_* is responsible for the metabolic pathway of 2,4-D biodegradation.

Atrazine and Cyanazine are both triazine-based herbicides. The genes involved in the biodegradation process of Atrazine are part of the *atz* operon, where *atzABC* genes are responsible for the transformation of Cyanazine to Cyanuric acid [374].

Understanding the genetics behind the biodegradation of pesticides is important for several reasons. Firstly, it allows for identifying and characterizing the genes and enzymes involved in pesticide degradation, which can help obtain more insights into the biochemical pathways and mechanisms of biodegradation. This knowledge can be used to develop bioremediation strategies and novel applications, such as the development of transgenic plants tolerant to herbicides.

Secondly, understanding the genetics of pesticide biodegradation can help in the assessment and monitoring of biodegradation processes in environmental settings, such as agricultural soils and bioremediation systems. This information is crucial for evaluating the efficiency and effectiveness of biodegradation processes and for designing strategies to mitigate environmental pollution.

Finally, studying the genetics of pesticide biodegradation can contribute to understanding microbial adaptation and evolution in response to selective pressures, such as organic xenobiotics. This knowledge can enhance our understanding of microbial ecology and the role of microorganisms in alleviating environmental pollution.

### 3.4. Application and Perspective

Pesticide bioremediation is a promising approach to mitigating the negative impacts of pesticides on the environment. Bioremediation can be achieved by using microorganisms such as fungi [375], bacteria [376], algae [279], and actinobacteria [377], all of them strong promising candidates to be used as bioremediating agents of pesticides. Through this process, different economically important chemicals, such as biofertilizers, biogas, or bioplastics [280,378], can be obtained.

Pesticide bioremediation has remained largely in the laboratory phase, where experiments under controlled conditions are performed. In order to be successfully used in situ, factors such as pesticide bioavailability, physiochemical conditions, temperature, pH, soil moisture, soil composition, surfactants, and organic amendments still reamain to be fully manageable [376].

When microorganisms are a constraint for the bioremediation process, enzymes [379] can be used. In an in situ scenario, free or immobilized enzymes are added to the contaminated soil or water, and degradation of the pollutant molecule is achieved through enzymatic activity. A successful pesticide bioremediation process using enzymes is dependent on several factors, the most important of which is enzyme stability.

New technologies could help achieve an effective pesticide biodegradation process. The use of nanoparticles to deliver pesticides is an alternative where a more precise quantity of pesticide is used, and due to the chemical and physical properties of the nanoparticles, biodegradation by the action of microorganisms could be more efficient [380,381,382]. Nanocarriers are also a potential alternative for pesticide biodegradation; biomolecules, such as enzymes, can be transported, attached to the nanoparticles, and delivered to the place where the pesticide is. Chitinases have been successfully immobilized in nanoparticles and tested for biocontrol against nematodes [383], opening the possibility of using nanocarriers for pesticide-degrading enzymes.

As the population increases, so does the production of crops; the necessity to maximize the production of these crops to meet the needs of the population will likely continue to be the implementation of pesticides. More research is needed for the development of different approaches and new technology, and their effective adoption is and will continue to be crucial for pesticide biodegradation.

## Figures and Tables

**Figure 1 ijms-24-15969-f001:**
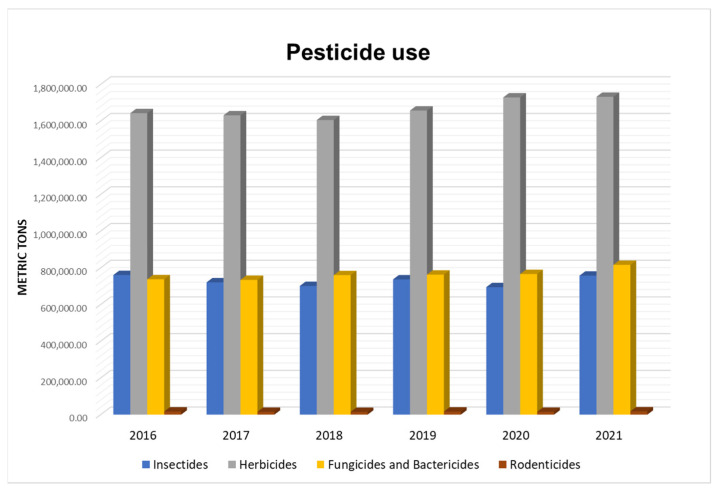
Pesticide use from 2016 to 2021 (data from https://www.fao.org/faostat, accessed on 18 October 2023).

**Figure 2 ijms-24-15969-f002:**
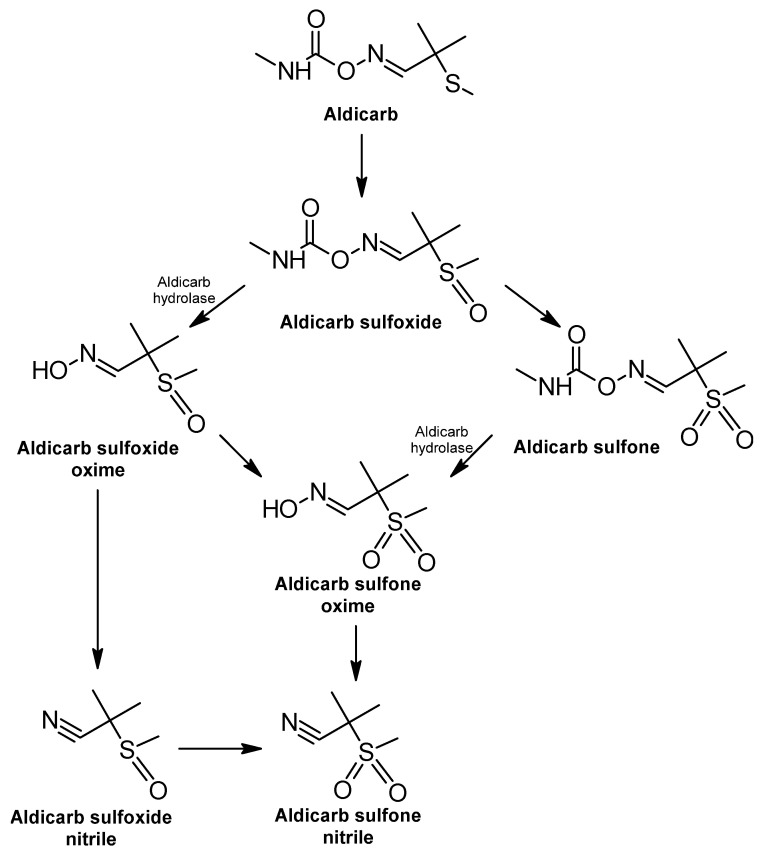
Aldicarb degradation pathway—adapted and re-drawn from [303].

**Figure 3 ijms-24-15969-f003:**
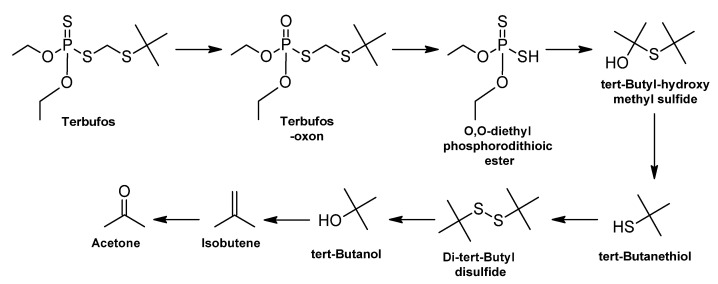
Terbufos biodegradation: metabolic pathway—adapted and re-drawn from [310].

**Figure 4 ijms-24-15969-f004:**
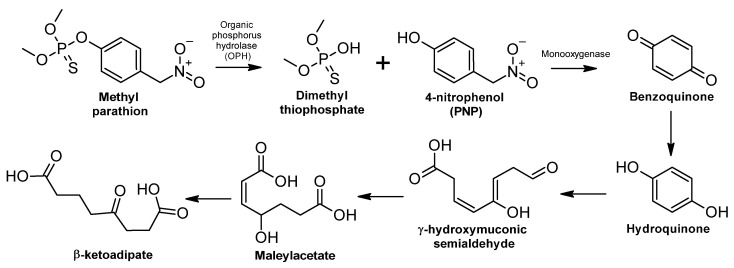
Methyl parathion biodegradation: metabolic pathway—adapted and re-drawn from [314].

**Figure 5 ijms-24-15969-f005:**
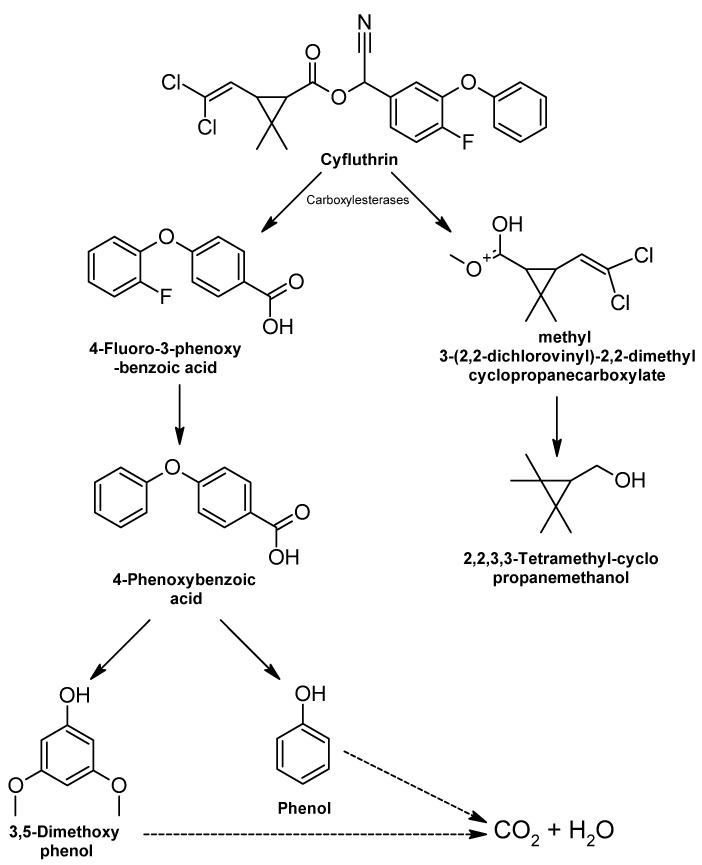
Cyfluthrin biodegradation: metabolic pathway—adapted and re-drawn from [88].

**Figure 6 ijms-24-15969-f006:**
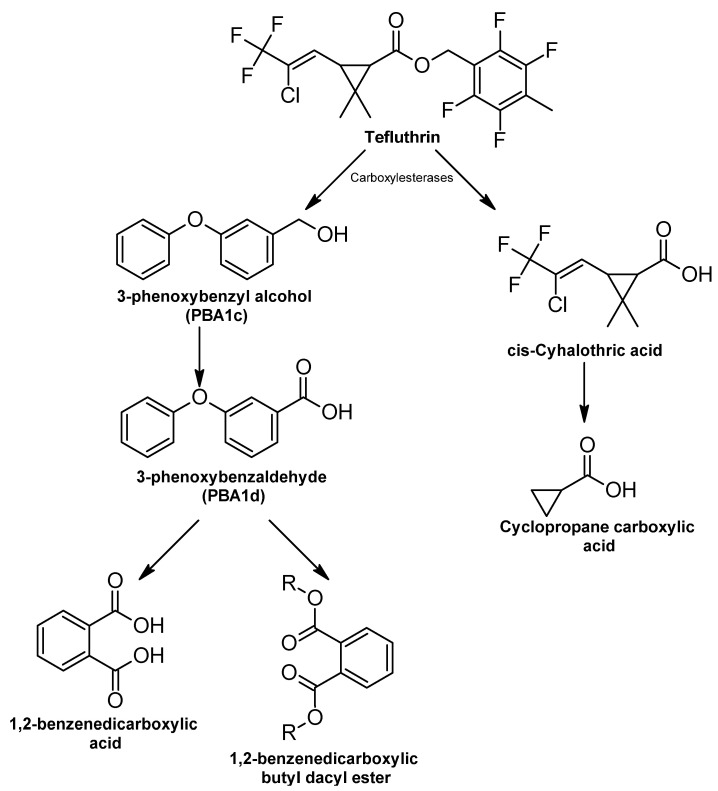
Tefluthrin degradation pathways—adapted and re-drawn from [25].

**Figure 7 ijms-24-15969-f007:**
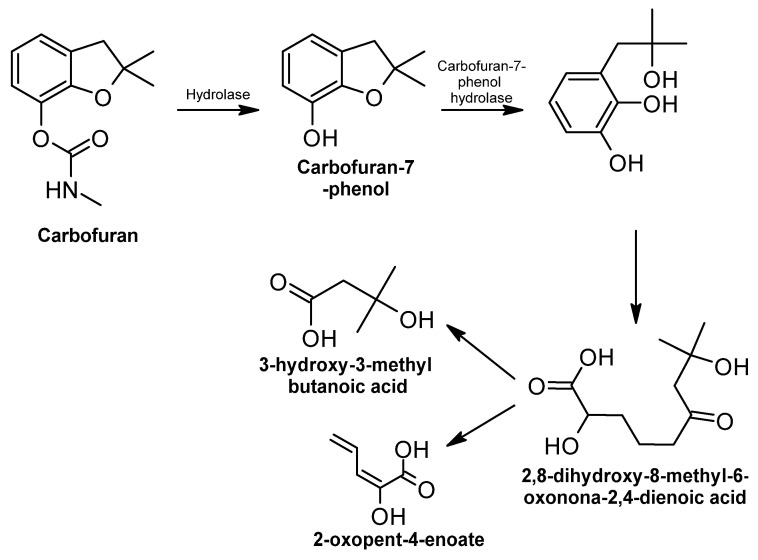
Carbofuran degradation pathway—adapted and re-drawn from [326].

**Figure 8 ijms-24-15969-f008:**
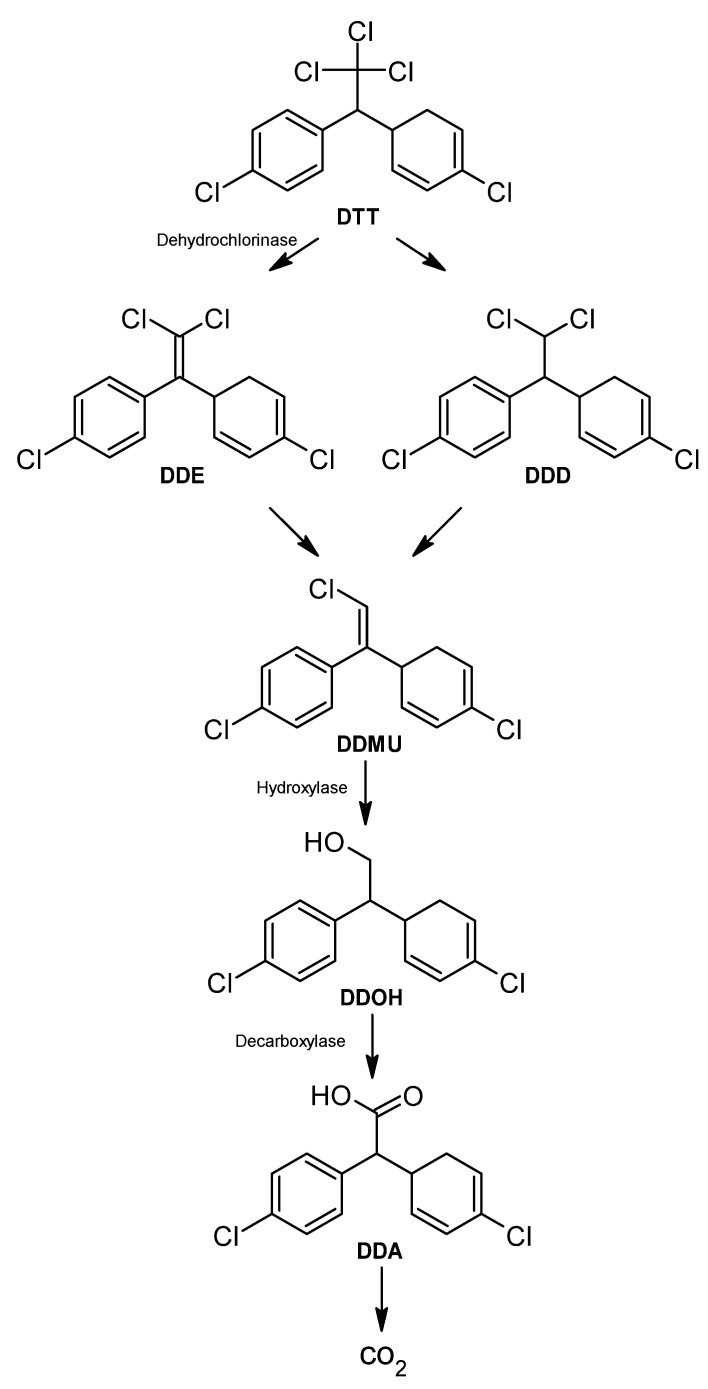
DDT degradation pathway—adapted and re-drawn from [98].

**Figure 9 ijms-24-15969-f009:**
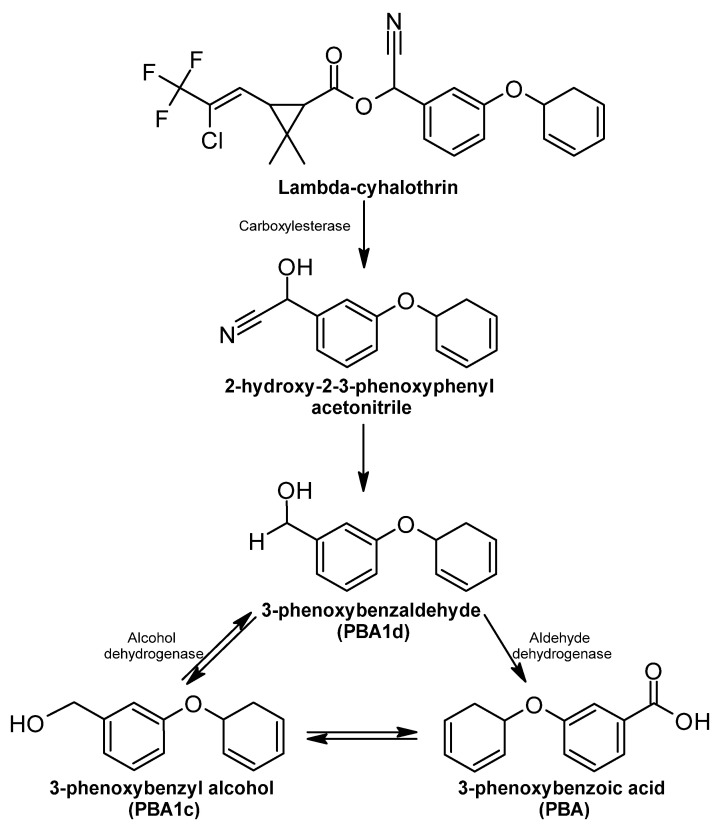
Lambda-cyhalothrin degradation pathway—adapted and re-drawn from [25].

**Figure 10 ijms-24-15969-f010:**
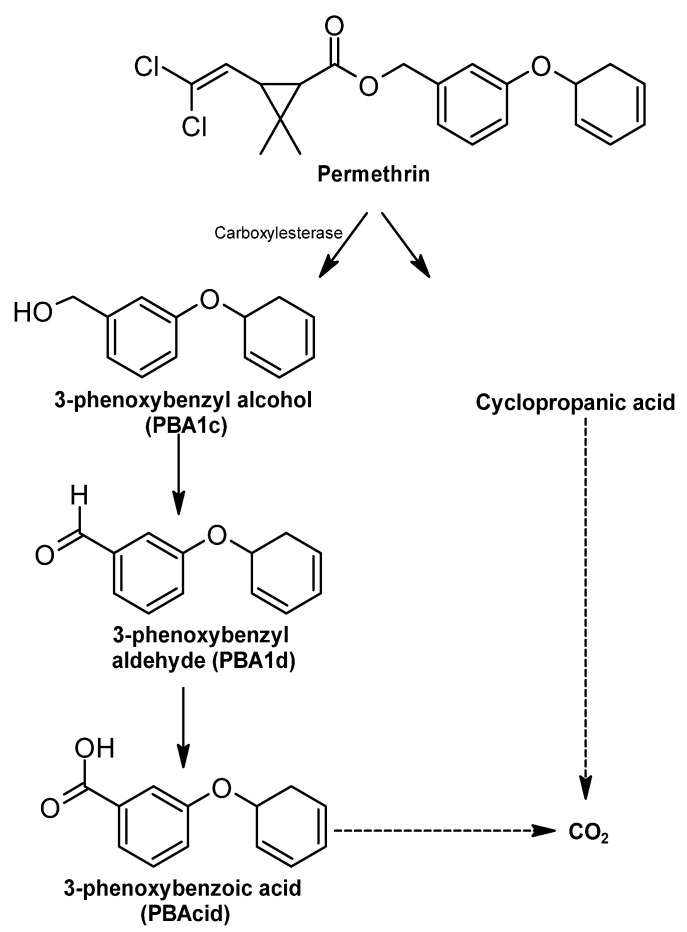
Permethrin degradation pathway—adapted and re-drawn from [343].

**Figure 11 ijms-24-15969-f011:**
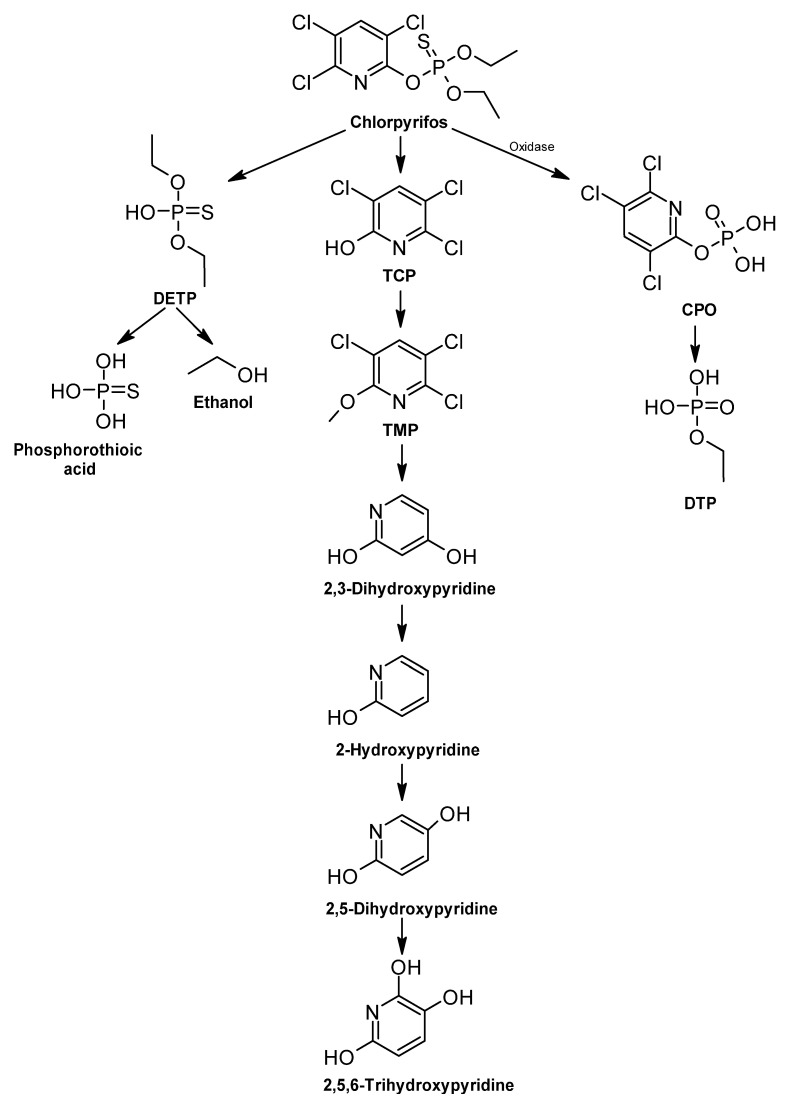
Chlorpyrifos degradation pathway—adapted and re-drawn from [346].

**Figure 12 ijms-24-15969-f012:**
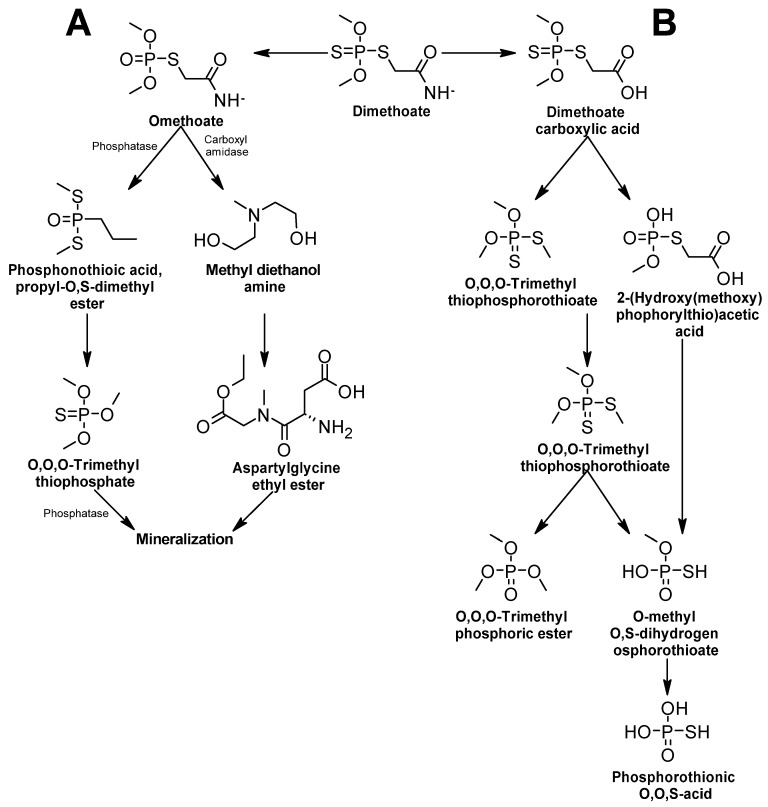
Dimethoate degradation pathway—adapted and re-drawn from [93,136]. (**A**) Dimethoate degradation via omethoate. (**B**) Dimethoate degradation via carboxylation.

**Figure 13 ijms-24-15969-f013:**
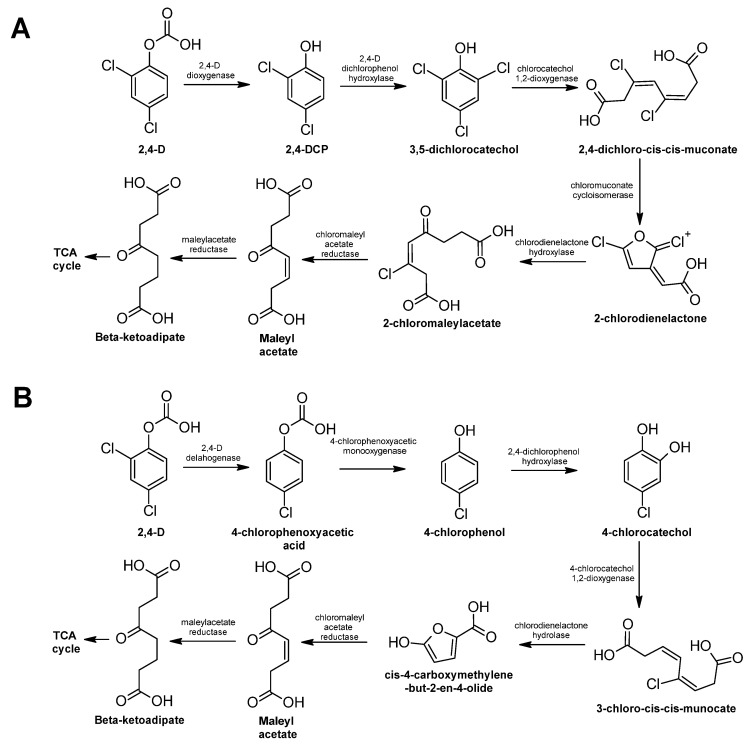
2,4-D degradation pathway—adapted and re-drawn from [349]. (**A**) 2,4-D biodegradation pathway dependent of the enzyme 2,4-D dioxygenase. (**B**) 2,4-D biodegradation pathway dependent on the enzyme 2,4-D dehalogenase.

**Figure 14 ijms-24-15969-f014:**
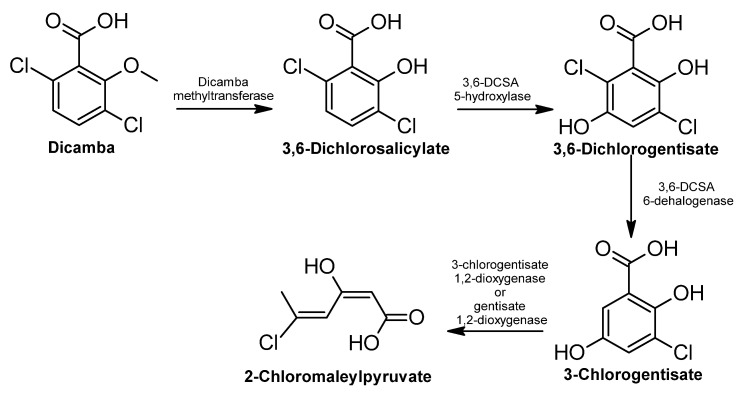
Dicamba degradation pathway—adapted and re-drawn from [357].

**Figure 15 ijms-24-15969-f015:**
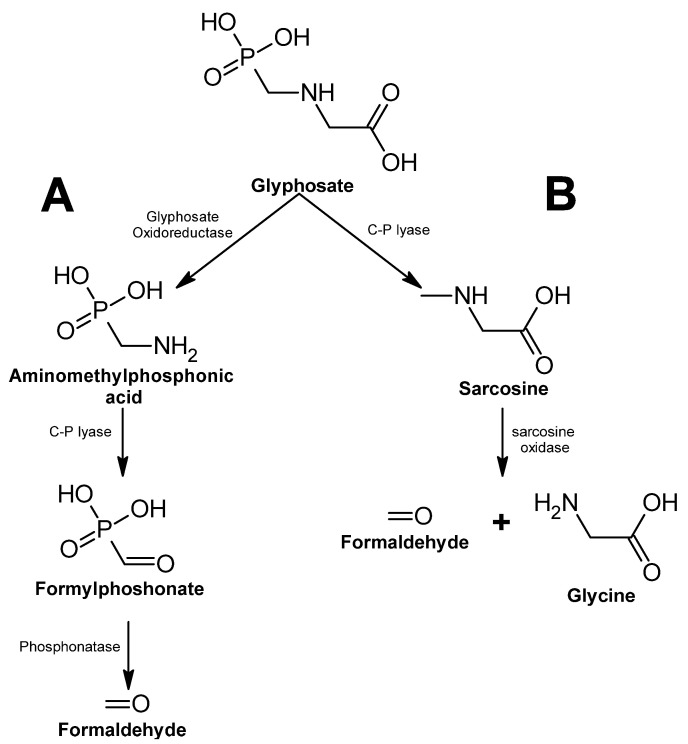
Glyphosate degradation pathway—adapted and re-drawn from [359]. (**A**) Glyphosate degradation pathway mediated by the enzyme Glyphosate Oxidoreductase. (**B**) Glyphosate degradation pathway mediated by the enzyme C-P lyase.

**Figure 16 ijms-24-15969-f016:**
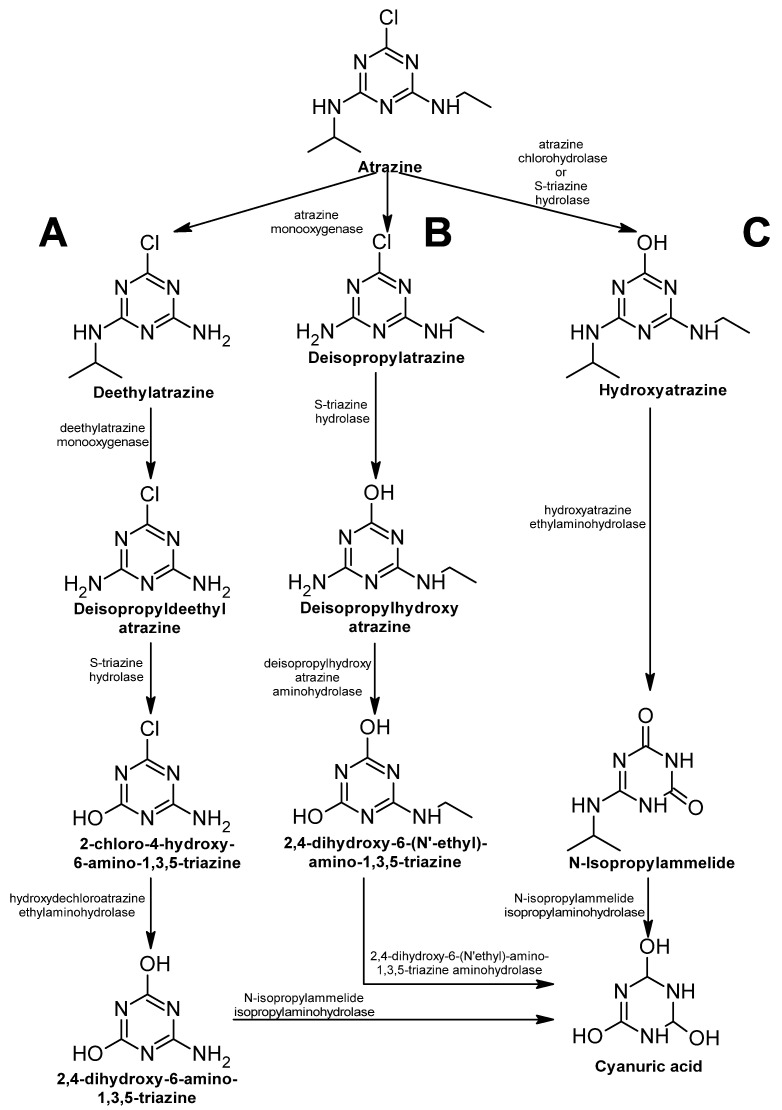
Atrazine degradation pathway—adapted and re-drawn from [360]. (**A**) Most common bacterial degradation pathway. (**B**,**C**) degradation pathways usually associated with microbial consortium.

**Figure 17 ijms-24-15969-f017:**
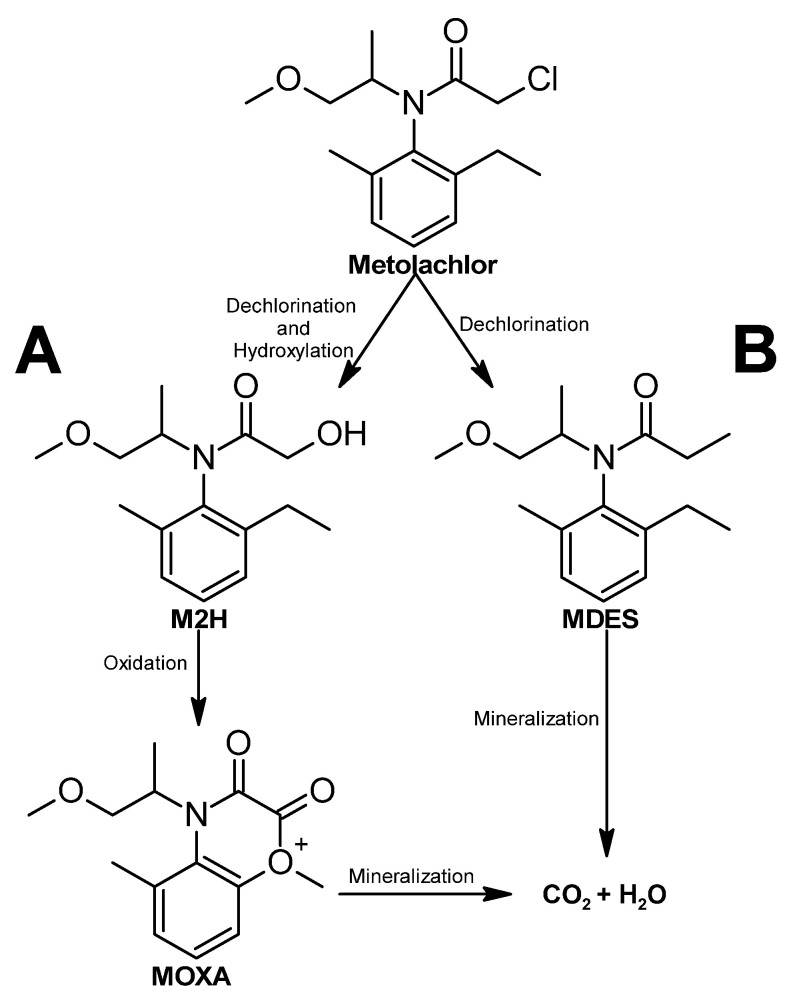
Metolachlor degradation pathway—adapted and re-drawn from [275]. (**A**) Metabolic pathway where hydroxylation is the first step. (**B**) Metabolic pathway where dechlorination is the first step.

**Table 1 ijms-24-15969-t001:** WHO pesticide classification.

Class	Oral LD_50_ for the Rat (mg/kg Body Weight)	Dermal LD_50_ for the Rat (mg/kg Body Weight)
Ia: Extremely hazardous	<5	<50
Ib: Highly hazardous	5–50	50–200
II: Moderately hazardous	50–2000	200–2000
III: Slightly hazardous	Over 2000	
U: Unlikely to present acute hazard	5000 or higher	

**Table 2 ijms-24-15969-t002:** Extremely hazardous pesticides used in agriculture.

Substance	LD_50_ in Rats (mg/kg Body Weight)	Reference
Aldicarb (Carbamate)	0.46–0.93	[12]
Terbufos (Organophosphate)	1.6–4.5	[13]
Methyl Parathion (Organophosphate)	6.9	[14]

**Table 3 ijms-24-15969-t003:** Highly hazardous pesticides used in agriculture.

Substance	LD_50_ in Rats (mg/kg Body Weight)	Reference
Cyfluthrin (Pyrethroid)	900	[21]
Tefluthrin (Pyrethroid)	21.8	[22]
Carbofuran (Carbamate)	7	[22]

**Table 4 ijms-24-15969-t004:** Moderately hazardous pesticides used in agriculture.

Substance	LD_50_ in Rats (mg/kg Body Weight)	Reference
DDT (Organochlorine)	113–118	[29]
λ-cyhalothrin (Pyrethroid)	612	[30]
Permethrin (Pyrethroid)	430–4000	[31]
Chlorpyrifos (Organophosphate)	135	[32]
Dimethoate (Organophosphate)	245	[33]
2,4-D (Organochlorine)	375	[34]
Dicamba (Organochlorine)	1581	[35]
Cyanazine (Organochlorine)	140–750	[36]

**Table 5 ijms-24-15969-t005:** Slightly hazardous pesticides used in agriculture.

Substance	LD_50_ in Rats (mg/kg Body Weight)	Reference
Glyphosate (Organophosphate)	7203.58–7397.25	[52]
Atrazine (Organochlorine)	1869	[53]
Metolachlor (Organochlorine)	1500	[54]

**Table 6 ijms-24-15969-t006:** Unlikely hazardous pesticides used in agriculture.

Substance	LD_50_ in Rats (mg/kg Body Weight)	Reference
Trifluralin (Organophosphate)	10,000	[59]

**Table 7 ijms-24-15969-t007:** Pesticide degrading bacteria.

WHO Pesticide Classification	Pesticide (% of Biodegradation Rate)	Bacteria	Reference
Extremely Hazardous	Aldicarb (85%) Terbufos (42%) Methyl parathion (7–90%)	*Enterobacter cloacae* TA7 *Micrococcus arborescens* *Pseudomonas aeruginosa* *Brachybacterium* sp. *Salsuginibacillus kocurii* *Stenotrophomonas* sp. YC-1 *Flavobacterium* sp. *Ochrobactrum* sp. B2 *Agrobacterium* sp. YW12 *Fischerella* sp. *Serratia* sp. DS001 *Bacillus* sp. CBMAI 1833 *Bacillus cereus* P5CNB *Pseudomonas* sp. Z1 *Burkholderia zhejiangensis* CEIB S4–3 *Nodularia linckia* *Nostoc muscorum* *Oscilatoria animalis* *Phormidium foveolarum* *Burkholderia cenocepacia* CEIB S5-2 *Acinetobacter* sp. *Pseudomonas putida* *Bacillus* sp. *Citrobacter freundii* *Stenotrophomonas* sp. *Flavobacterium* sp. *Proteus vulgaris* *Pseudomonas* sp. *Acinetobacter* sp. *Klebsiella* sp. *Proteus* sp. *Microcystis novacekii* *Alcaligenes* sp. SRG *Serratia marcescens* MEW06	[20,67,68,69,70,71,72,73,74,75,76,77,78,79,80,81,82,83]
Highly Hazardous	Cyfluthrin (80%) Tefluthrin ^1^ Carbofuran (97.5%)	*Photobacterium ganghwense* T14 *Novosphingobium* sp. KN65.2 *Pseudomonas stutzeri* S1 *Lysinibacillus sphaericus* FLQ-11-1 *Brevibacterium aureum* *Cupriavidus* sp. ISTL7 *Enterobacter cloacae* TA7 *Bacillus* sp. *Chryseobacterium* sp. BSC2-3 *Burkholderia cepacia* PCL3	[67,84,85,86,87,88,89,90,91,92]
Moderately Hazardous	DDT (5–98%) Lambda- Cyhalothrin (70–90%) Permethrin (80–100%) Chlorpyrifos (60–90%) Dimethoate (80–98%) 2,4-D (30–90%) Dicamba ^1^ Cyanazine ^1^	*Alcaligenes faecalis* strain DSP3 *Pseudomonas nitroreducens* AR-3 *Ralstonia pickettii* *Stenotrophomonas* sp. *Pseudomonas aeruginosa* *Ochrobactrum* sp. DDT-2 *Alcaligenes* sp. KK *Arthrobacter globiformis* DC-1 *Serratia marcescens* NCIM 2919 *Advenella kashmirensis* *Corynebacterium* sp. *Enterobacter cloacae* *Bacillus thuringiensis* ZS-19 *Bacillus velezensis* sd *Bacillus subtilis* *Paracoccus acridae* SCU-M53 *Brucella intermedia* Halo1 *Alcaligenes faecalis* CH1S *Aquamicrobium terrae* CH1T *Enterobacter ludwigii* *Bacillus thuringiensis* Berliner *Acinetobacter baumannii* ZH-14 *Hortaea* sp. B15 *Streptomyces* sp. *Enterobacter* sp. SWLC2 *Pseudomonas putida* CBF10-2 *Ochrobactrum anthropi* FRAF13 *Rhizobium radiobacter* GHKF11 *Methylobacterium extorquens* *Bacillus cereus* Ct3 *Stenotrophomonas maltophilia* *Acinetobacter calcoaceticus* *Bacillus amyloliquefaciens* CP28 *Pseudomonas putida* T7 *Pseudomonas aeruginosa* M2 *Klebsiella pneumoniae* M6 *Alcaligenes* sp. *Bacillus subtilis* *Enterobacter* sp. *Klebsiella* sp. *Micrococci* sp. *Cupriavidus nantongensis* X1T *Bacillus megaterium* CCLP1 *Bacillus safensis* CCLP2 *Shewanella* sp. BT05 *Pseudomonas fluorescens* CD5 *Achromobacter spanius* C1 *Pseudomonas rhodesiae* C4 *Weissella confusa* *Azotobacter vinelandii* ATCC 12837 *Coleofasciculus chthonoplastes* *Lysinibacillus* sp. HBUM206408 *Achromobacter insuavis* *Dyadobacter jiangsuensis* 12851 *Arthrobacter* sp. HM01 *Psychrobacter alimentarius* T14 *Streptomyces phaeochromogenes* *Streptomyces praecox* SP1 *Pseudomonas putida* *Xanthomonas campestris pv. Translucens* *Pseudomonas kilonensis* MB490 *Serratia* sp. (100%) *Sphingomonas* sp. DC-6 *Raoultella* sp. X1 *Lactobacillus plantarum* *Chryseobacterium* *Variovorax* *Aeromonas* *Xanthobacter* *Acidovorax* *Cupriavidus gilardii* T-1 *Enterobacter hormaechei* subsp. xiangfangensis 19_357_F *Cupriavidus campinensis* *Delftia* sp. *Cupriavidus necator* *Arthrobacter* sp. SVMIICT25 *Sphingomonas* sp. SVMIICT11 *Stenotrophomonas* sp. SVMIICT13 *Corynebacterium humireducens* MFC-5 *Cupriavidus oxalaticus* X32 *Thauera* sp. DKT *Pseudomonas simiae* EGD-AQ6 *Sphingobium* sp. Ndbn-10 *Sphingomonas* sp. Ndbn-20 *Pseudomonas maltophilia* DI-6 *Rhizorhabdus dicambivorans* Ndbn-20	[37,93,94,95,96,97,98,99,100,101,102,103,104,105,106,107,108,109,110,111,112,113,114,115,116,117,118,119,120,121,122,123,124,125,126,127,128,129,130,131,132,133,134,135,136,137,138,139,140,141,142,143,144,145,146,147,148,149,150,151,152,153,154,155]
Slightly Hazardous	Glyphosate (30–90%) Atrazine (40–100%) Metolachlor (40–100%)	*Rhodococcus soli* G41 *Stenotrophomonas maltophilia* GP-1 *Achromobacter* sp. MPK7A *Bacillus cereus* *Burkholderia vietnamiensis* AO5–12 *Burkholderia* sp. AO5–13 *Enterobacter cloacae* K7 *Ochrobacterium anthropic* GPK3 *Pseudomonas* sp. *Agrobacterium tumefaciens* CNI28 *Novosphingobium* sp. CNI35 *Ochrobactrum pituitosum* CNI52 *Gallinifaecis* sp. CAS4 *Chryseobacterium* sp. Y16C *Spirulina platensis* *Streptomyces lusitanus* *Lysinibacillus sphaericus* *Stenotrophomonas acidaminiphila* Y4B *Bradyrhizobium* spp. *Comamonas odontotermitis* P2 *Bacillus subtilis* *Rhizobium leguminosarum* *Serratia* sp. *Bacillus aryabhattai* FACU *Pseudomonas fluorescens* *Providencia rettgeri* GDB 1 *Bacillus cereus* *Pseudomonas alcaligenes* *Pseudomonas stutzeri* *Bacillus licheniformis* *Lactobacillus plantarum* *Lactobacillus rhamnosus* *Bacillus shackletonii* *Pseudomonas citronellolis* ADA-23B *Solicoccozyma terricola* M 3.1.4. *Achromobacter denitrificans* *Ochrobactrum haematophilum* *Pseudomonas putida* Ch2 *Ochrobactrum intermedium* Sq20 *Burkholderia cepacia* PSBB1 *Pseudomonas aeruginosa* *Ensifer adhaerens* SZMC 25856 *Pseudomonas resinovorans* SZMC 25875 *Burkholderia anthina* *Burkholderia cenocepacia* *Geobacillus caldoxylosilyticus* T20 *Bacillus licheniformis* ATLJ-5. *Bacillus megaterium* ATLJ-11 *Pseudomonas* sp. *Pseudaminobacter* sp. *Nocardioides* sp. *Klebsiella* sp. A1 *Comamonas* sp. A2 *Klebsiella variicola* FH-1 *Arthrobacter* sp. NJ-1 *Acinetobacter lwoffii* DNS32 *Agrobacterium rhizogenes* AT13 *Acetobacter* *Pseudomonas* *Clostridium-sensu-stricto* *Burkholderia* *Ensifer* sp. *Solibacillus* sp. *Bacillus* sp. *Arthrobacter* sp. *Bacillus velezensis* MHNK1 *Citricoccus* sp. TT3 *Paenarthrobacter* sp. W11 *Methylobacillus* *Enterobacter* sp. P1 *Arthrobacter* sp. DNS10 *Bradyrhizobium* *Rhodococcus* *Hydrogenophaga* sp. Gsoil 1545 *Sinorhizobium* sp. TJ170 *Rhizobium* sp. *Bacillus anthracis* *Pseudomonas balearica* *Pseudomonas otitidis* *Pseudomonas indica* *Providencia vermicola* *Pseudomonas* spp. ACB *Pseudomonas* spp. TLB *Methylobacterium* *Mycobacterium* *Bacillus atrophaeus* *Variovorax* sp. 38R *Arthrobacter* sp. TES *Chelatobacter* sp. SR38 *Bacillus megaterium* Mes11 *Ralstonia* *Phyllobacterium* *Stenotrophomonas* *Holophaga foetida*	[55,156,157,158,159,160,161,162,163,164,165,166,167,168,169,170,171,172,173,174,175,176,177,178,179,180,181,182,183,184,185,186,187,188,189,190,191,192,193,194,195,196,197,198,199,200,201,202,203,204,205,206,207,208]
Unlikely Hazardous	Trifuralin (30–95%)	*Arthrobacter aurescens CTFL7* *Herbaspirillum* sp. *Klebsiella* sp. *Pseudomonas fluorescens* *Bacillus simplex* *Bacillus muralis* *Micrococcus luteus* *Micrococcus yunnanensis* *Clostridium tetani* *Klebsiella oxytoca* *Herbaspirillum seropedicae* *Bacillus megaterium* *Brevundimonas diminuta* *Streptomyces* PS1/5	[123,209,210,211,212,213,214]

^1^ Biodegradation rates for these pesticides are not reported.

## Data Availability

Not applicable.

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
