# Peer review of "Microbiology and Biochemistry of Pesticides Biodegradation"

_ijms, 2023, doi:10.3390/ijms242115969_

Round 1
Reviewer 1 Report
The review article by Guerrero Ramírez et al. is a study on the biodegradation of pesticides. Here are some comments to improve the article, which, while interesting, seems rather superficial (perhaps the focus should be on a more specific group of pesticides):
1) The abstract is more a form of introduction to the subject matter, but does not provide insight into what issues are described in the publication.
2) Please note minor editorial errors, such as the lack of a lower index in LD50.
3) Only a few representatives from each risk group are described in Chapter 2. Why such a choice? What was the reasoning behind it? Shouldn't this section be expanded and at least list more representatives of each category?
4) Chapter 3 also refers to only selected compounds. Why exactly were these described?
5) The drawings are of relatively poor quality, I suggest redrawing them.
6) In order to obtain more general observations, perhaps the work should be divided not by risk level, but by the chemical groups to which the individual compounds belong, perhaps then it would be easier to observe which strains and which enzymes/gens participate in biodegradation.
7) There is no information in the paper about biodegradation efficiency, toxicity to the microorganisms that biodegrade them - this should be completed.
8) How about focusing on bio-degradation by bacteria only or fungi only or algae only? Then a given topic would be easier to describe more comprehensively.
Author Response
1) The abstract is more a form of introduction to the subject matter, but does not provide insight into what issues are described in the publication.
R= We would like to thank the reviewer for the observation regarding the abstract; the reviewer can read an expanded abstract in the new document.
2) Please note minor editorial errors, such as the lack of a lower index in LD50.
R= The authors thank the reviewer for the observation; all lower indexes have been properly added.
3) Only a few representatives from each risk group are described in Chapter 2. Why such a choice? What was the reasoning behind it? Shouldn't this section be expanded and at least list more representatives of each category?
R= The reasoning behind the selection of the representatives of each risk group is that although the WHO provides an LD50 for each pesticide included in a particular group (e.g., 29 pesticides in group Ia) the authors could retrieved scientific articles that confirmed the LD50 for only 3 of those 29 chemicals, not to say that the WHO is giving unsubstantiated information but rather that the experimental data that backs up the LD50 of pesticides is not entirely easily accessible for the public. This was the main filter that the authors used to select the pesticides that were discussed in the following sections.
4) Chapter 3 also refers to only selected compounds. Why exactly were these described?
R= Metabolic pathways were documented for the same pesticides selected in Section 2.
5) The drawings are of relatively poor quality; I suggest redrawing them.
R= We thank the reviewer for the observation, all images were redrawn and included in the new document.
6) In order to obtain more general observations, perhaps the work should be divided not by risk level, but by the chemical groups to which the individual compounds belong, perhaps then it would be easier to observe which strains and which enzymes/gens participate in biodegradation.
R= We thank the reviewer for the suggestion, although a chemical similarity classification could help to associate strains and enzymes that participate in biodegradation, we decided to use a risk level classification to emphasise that there are pesticides that present very low toxic effects to nontarget organisms that could be used in a higher degree, for example, both terbufos (extremely hazardous) and trifluralin (unlikely hazardous) are organophosphate pesticides with a similar action mechanism, but terbufos is preferred as a preemergence pesticide. Chemical classification was added to each of the pesticides.
7) There is no information in the paper on biodegradation efficiency, toxicity to the microorganisms that biodegrade them; this should be completed.
R= We thank the reviewer for the observation, a more comprehensive list of microorganisms can be found in the corrected document, and the biodegradation rate was added to each pesticide.
8) How about focusing on bio-degradation by bacteria only or fungi only or algae only? Then a given topic would be easier to describe more comprehensively.
R= We would like to thank the reviewer for the suggestion, we decided to include both fungi and algae in the manuscript because these two have been less studied than bacteria, for example, practically all the information we were able to retrieve about metabolic pathways came from bacteria. A discussion addressing this issue was added at the end of section 3.2.
Reviewer 2 Report
The reviewed article is a review on the biodegradation processes of selected pesticides commonly used for crop protection. The article is a very well prepared compendium of current knowledge on the possibilities and pathways of the biodegradation process of the pesticides analysed in the paper. The authors have carried out an extensive literature survey and described the results of recent research in this field. The article is very interesting. However, there are some deficiencies or editorial errors regarding the figures that make them difficult to understand.
1. Almost all the figures show several different metabolic pathways, but there is no information in the titles of the figures or in the text which microorganisms carry out which metabolic pathway.
2. Figure 12 is divided into two parts A and B, with no explanation of what these letters mean.
3. Figures 2, 3, 11, 12, 13 and 15 have too small a font and are hardly readable.
4. Figures 6, 7 and 10 are in a different format to the others or have been copied incorrectly (the rings look completely different).
5. The title of figure 3 is in the wrong format.
6. The subtitles 2.1.1-4 and 3.2.1-4 should be changed (word 'pesticides' should be added).
Author Response
The review article is a review on the biodegradation processes of selected pesticides commonly used for crop protection. The article is a very well prepared compendium of current knowledge on the possibilities and pathways of the biodegradation process of the pesticides analysed in the paper. The authors have carried out an extensive literature survey and described the results of recent research in this field. The article is very interesting. However, there are some deficiencies or editorial errors regarding the figures that make them difficult to understand.
1. Almost all the figures show several different metabolic pathways, but there is no information in the titles of the figures or in the text which microorganisms carry out which metabolic pathway.
2. Figure 12 is divided into parts A and B, with no explanation of what these letters mean.
3. Figures 2, 3, 11, 12, 13 and 15 have too small a font and are hardly readable.
R= We would like to thank the reviewer for the observation, all images were redrawn, and the format is the same for each one.
- Figures 6, 7 and 10 are in a different format to the others or have been copied incorrectly (the rings look completely different).
R= We would like to thank the reviewer for the observation, all images were redrawn, and the format is the same for each one.
- The title of figure 3 is in the wrong format.
R= We would like to thank the reviewer for the observation, and the title has been corrected.
- The subtitles 2.1.1-4 and 3.2.1-4 should be changed (the word 'pesticides' should be added).
R= We thank the reviewer for the observation, the word pesticides has been added to all the subtitles.
Reviewer 3 Report
The review is an attempt at an in-depth look at the microbial degradation of pesticides. Since the application of pesticides in agriculture is inevitable, and one approach to their degradation is through bioremediation, the article deserves attention. However, many parts of it need improvement and editing.
1. Introduction: there are not enough statistics on where pesticides are found: water, food, soil, or other sources. Where does their danger to man come from, and in what way is he exposed to them - through water, food, or air?
2. Chapter 2 does not contain any information. If you are going to write what pesticides are used in agriculture, please specify them: insecticides, herbicides, fungicides, against what pests, for what crops, and in what quantity. A table of these statistics for one of the recent years should be added to make the section informative.
3. The classification of pesticides depends on several indicators, in this case, the authors decided to classify them according to the degree of danger to humans. However, the examples in the tables are too scarce. With hundreds of pesticides available, please provide more examples. There is a single substance in Table 6, in which case consider whether the table should exist, or if the example can only be described in the text.
4. Since the authors consider the biological pathways of pesticide degradation, the "classification" section must necessarily include one according to the chemical nature of the pesticides. I propose a table in which to describe the classes of pesticides according to their chemical composition.
5. Section 3.1.1 omitted lactic acid bacteria with huge potential in food detoxification of a number of pesticides, please add them. Here are some examples:
https://pubmed.ncbi.nlm.nih.gov/36981090/, https://pubs.acs.org/doi/10.1021/acs.jafc.0c03948, https://academic.oup.com/lambio/article -abstract/57/5/412/6699501?redirectedFrom=fulltext.
6. Minor Notes:
Line 39: prevent is doubled.
The figures are not in the same style if the quality of Figure 7 (flattened) and Figure 14 (strongly different from the others in the size of compound formulas and arrows) could be improved. Color can be included in figures to highlight important enzymes or catalytic steps. The text under Figure 3 should not be in italics.
Minor editing of English language required.
Author Response
- Introduction: there are not enough statistics on where pesticides are found: water, food, soil, or other sources. Where does their danger to man come from, and in what way is he exposed to them - through water, food, or air?
R= We would like to thank the reviewer for the observation, as we would very much like to present that type of data; the statistics of pesticide contamination/exposure are not available, at least not to the best of the authors knowledge. What can be found in the text is information about detection in either water, air or soil for specific pesticides, for example, for aldicarb (page 4), carbofuran (page 4), DDT (page 5), lambda-cyhalothrin (page 5), Permethrin (page 5), cyanazine (page 5), Glyphosate (page 6).
- Chapter 2 does not contain any information. If you are going to write what pesticides are used in agriculture, please specify them: insecticides, herbicides, fungicides, against what pests, for what crops, and in what quantity. A table of these statistics for one of the recent years should be added to make the section informative.
R= We would like to thank the reviewer for the suggestion, a figure with the number of metric tons used from 2016 to 2021 for each type of pesticide has been added to Section 2.
- The classification of pesticides depends on several indicators, in this case the authors decided to classify them according to the degree of danger to humans. However, the examples in the tables are too scarce. With hundreds of pesticides available, provide more examples. There is a single substance in Table 6, in which case consider whether the table should exist or if the example can only be described in the text.
R= We would like to thank the reviewer for the observation; the reasoning behind the selection of the representatives of each risk group is that although the WHO provides an LD50 for each pesticide included in a particular group (eg, 29 pesticides in group Ia) the authors could retrieve scientific articles that confirmed the LD50 for only 3 of those 29 chemicals, not to say that the WHO is giving unsubstantiated information but rather that the experimental data that support the LD50 of pesticides is not entirely easily accessible for the public. This was the main filter that the authors used to select the pesticides that were discussed in the following sections.
- Since the authors consider the biological pathways of pesticide degradation, the "classification" section must necessarily include one according to the chemical nature of the pesticides. I propose a table in which to describe the classes of pesticides according to their chemical composition.
R= We thank the reviewer for the observation, and chemical classification was added to each one of the pesticides.
- Section 3.1.1 omitted lactic acid bacteria with a huge potential in food detoxification of a number of pesticides; please add them. Here are some examples:
https://pubmed.ncbi.nlm.nih.gov/36981090/, https://pubs.acs.org/doi/10.1021/acs.jafc.0c03948, https://academic.oup.com/lambio/article -abstract/57/5/412/6699501?redirectedFrom=fulltext.
R= We would like to thank the reviewer for the observation; a more comprehensive list of microorganisms can be found in the corrected document.
- Minor Notes:
Line 39: prevent is doubled.
Figures are not in the same style if the quality of Figure 7 (flattened) and Figure 14 (strongly different from the others in the size of compound formulas and arrows) could be improved. Colour can be included in the figures to highlight important enzymes or catalytic steps. The text in Figure 3 should not be in italics.
R= We thank the reviewer for the observation, all format errors have been corrected in the new document.
Round 2
Reviewer 1 Report
Thank you for your responses to my comments. I think the manuscript has been significantly improved.
Reviewer 3 Report
In general, the authors have answered the questions and met the requirements. The figures have been fixed. The manuscript may be accepted in its current form.
Minor editing of English language required.